# Improving Multi-Class Calibration through Normalization-Aware Isotonic Techniques

**Alon Arad** [1]    **Saharon Rosset** [1]

## Abstract

Accurate and reliable probability predictions are essential for multi-class supervised learning tasks, where well-calibrated models enable rational decision-making. While isotonic regression has proven effective for binary calibration, its extension to multi-class problems via one-vs-rest calibration produced suboptimal results when compared to parametric methods, limiting its practical adoption. In this work, we propose novel isotonic normalization-aware techniques for multi-class calibration, grounded in natural and intuitive assumptions expected by practitioners. Unlike prior approaches, our methods inherently account for probability normalization by either incorporating normalization directly into the optimization process (**NA-FIR**) or modeling the problem as a cumulative bivariate isotonic regression (**SCIR**). Empirical evaluation on a variety of text and image classification datasets across different model architectures reveals that our approach consistently improves negative log-likelihood (NLL) and expected calibration error (ECE) metrics.

## 1. Introduction

The rapid progress of artificial intelligence (AI) has greatly expanded both the trust in and the range of applications for AI models in real-world settings. These advancements have fundamentally reshaped fields requiring accurate predictions and informed decision-making, including healthcare, agriculture, finance, and retail. However, decision-making in these contexts often transcends simple notions of correctness, requiring a nuanced understanding of the underlying probabilities generated by the models. For practitioners, this means engaging critically with model outputs, assessing their reliability and implications within the broader decision-making framework. For instance, in healthcare, overestimating the likelihood of a disease may lead to unnecessary interventions, while underestimation risks delaying vital treatments.

Calibration is a term frequently used to describe this concept and serves a dual purpose in addressing these challenges. It is both a measure of the "reliability" of model outputs—evaluating the alignment between predicted probabilities and observed outcomes—and as a method for refining these outputs to enhance their practical utility. The concept of calibration has a long history, with foundational works (Murphy, 1973; Murphy & Winkler, 1977; DeGroot & Fienberg, 1981) emphasizing the relationship between loss scores and calibration. Subsequent research (Platt, 1999; Zadrozny & Elkan, 2001; Niculescu-Mizil & Caruana, 2005) has examined the reliability of outputs from various machine learning algorithms and the application of post-hoc calibration methods, particularly focusing on methods designed for binary classification tasks. To address the challenges of multi-class calibration, it was suggested (Zadrozny & Elkan, 2002) to decompose the problem into multiple binary classification tasks learnt only with class respective model output and binary label. To ensure that the predictions sum to 1 and form valid probability vector, the post-calibration predicted class values are normalized by dividing each by their sum.

Recent advancements in computational power and neural network architectures have introduced new challenges around over-confident models (Guo et al., 2017; Kull et al., 2019), particularly in quantifying and defining calibration in multi-class settings (Vaicenavicius et al., 2019). These works have highlighted the effectiveness of "simple" parametric approaches, such as Temperature Scaling, which naturally extend to multi-class scenarios without requiring decomposition binary sub-problems or additional normalization considerations, leaving traditional non-parametric methods somewhat sidelined.

However, later studies (Gupta & Ramdas, 2022; Patel et al., 2020) have underscored the untapped potential of traditional techniques when applied with well suited assumptions that have proven useful on empirical datasets evaluation. No-

*Equal contribution [1]Department of Statistics, Tel Aviv University. Correspondence to: Alon Arad <alonarad1.mail.tau.ac.il>, Saharon Rosset <saharon@tauex.tau.ac.il>.

*Proceedings of the 42nd International Conference on Machine Learning*, Vancouver, Canada. PMLR 267, 2025. Copyright 2025 by the author(s).

tably, these works focus on histogram binning (HB), despite the historical prominence of isotonic regression (IR) in binary classification tasks. Interestingly, both report state-of-the-art (SOTA) results for unnormalized predictions, meaning the predictions don't necessarily form a valid probability vector. This appears to paradoxically conflict with the fundamental goal of calibration: fostering trust in the reported probabilities.

In this work, we emphasize the importance of being *Normalization-Aware* in non-parametric multi-class settings. Sections 2 and 3 provide a concise overview of the subject and prior literature. In Section 4 we introduce two IR based approaches that incorporate normalization directly into the problem formulation and are motivated by simply stated assumptions. The first approach modifies the optimization function explicitly (hence the term Normalization-Aware), while the second redefines the problem by addressing cumulative sub-problems instead of treating each class as an independent binary task. As demonstrated in Section 5, our proposed methods consistently improve both negative log-likelihood (NLL) and calibration error across diverse datasets, achieving SOTA results and reaffirming the effectiveness of IR for calibration. We believe this provides practitioners with a powerful non-parametric alternative in scenarios where parametric assumptions may be limiting.

## 2. Problem Definition

### 2.1. Introduction to the Calibration Challenge

We address supervised multi-class classification where a classifier $\hat{p}$ learns to map inputs $X \in \mathbb{R}^d$ to finite set of classes $\{1, \ldots, k\}$ using a training set $\mathcal{T} = \{(x_i, y_i)\}$ where each pair $(x_i, y_i)$ represents a realization of i.i.d random variables $(X, Y)$, with $Y$ one-hot encoded vector. The classifier outputs predictions $\hat{p}(X)$ that lie within the probability simplex $\Delta^{k-1}$, defined as $\Delta^{k-1} = \left\{ \mathbf{p} \in \mathbb{R}^k \mid p_i \geq 0 \ \forall i, \ \sum_{i=1}^{k} p_i = 1 \right\}$. In essence, calibration implies that if a set of samples is classified with an 80% probability of belonging to a certain class, we would expect that, indeed, 80% of those samples accurately fall into that class. This requirement is formalized as follows:

**Definition 2.1.** A classifier $\hat{p}$ is called **calibrated** if for any $q \in \Delta^{k-1}$ it holds that

$$\mathbb{E}(Y \mid \hat{p}(X) = q) = q$$

While this definition effectively captures the theoretical essence of calibration, its measurement is often impractical in multi-class scenarios ($k > 4$) due to the extensive number of test points required for estimating it effectively. To address this complexity, a more attainable standard was proposed:

**Definition 2.2.** A classifier $\hat{p}$ is **class-wise calibrated** if

$$\mathbb{E}\big[Y_j \mid \hat{p}(X)_j\big] = \hat{p}(X)_j, \quad \forall j \in \{1, \ldots, k\}.$$

As demonstrated by Vaicenavicius et al. (2019) being **class-wise calibrated** does not necessarily imply that the classifier is indeed **calibrated**, making it a weaker concept of calibration. Additionally, another weaker notion of calibration proposed by Guo et al. (2017) has gained attention and popularity due to its strong practical motivation for addressing the question: *"Can we trust the top predicted category score?"* It is defined as follows:

**Definition 2.3.** a classifier $\hat{p}$ is **confidence calibrated** if

$$\mathbb{E}\big[Y_{\arg\max \hat{p}(X)} \mid \max \hat{p}(X)\big] = \max \hat{p}(X)$$

Offering a practical and interpretable metric for trust in classification outputs.

### 2.2. Error Measurement

A commonly used method for assessing the alignment of probabilistic binary forecasts with actual outcome probabilities is using a Reliability plot (see Murphy & Winkler 1977; DeGroot & Fienberg 1983; Niculescu-Mizil & Caruana 2005) This approach involves dividing predictions within $[0, 1]$ range into equal-width bins and comparing the model's predicted probabilities against the actual outcome statistics within each bin. While Reliability plot offers valuable visualization, single summary statistic is often preferred. In the binary case Naeini et al. (2015) define estimated calibration error (ECE) as the weighted sum of absolute differences in these bins. However, generalizing this measure is not straightforward, thus several definitions were offered. Nixon et al. (2019) suggested **cw-ECE** corresponding to the class-wise calibrated notion in Definition 2.2 and is estimated by averaging ECE computed independently for each class binary subproblem. Since this approach can overemphasize observations in small bins, they also propose **TECE**, which applies thresholded equal-mass binning to each class before averaging the resulting estimates. For the confidence calibrated notion in Definition 2.3, the **conf-ECE** estimator (Guo et al., 2017; Kull et al., 2019) is computed by applying binning and binary labeling based on the maximum predicted probability. The corresponding Reliability plot is known as Confidence Reliability plot. Alternative methods based on kernel density estimation (KDE) have been proposed (Zhang et al., 2020; Widmann et al., 2019; Marx et al., 2024) but are less frequently used in practice.

**Proper Scoring Rules.** Evaluation measures for probabilistic forecasts can be attributed back to Brier (1950) and Good (1952) where the main concern was creating a scoring rule to encourage the forecaster to be honest (see Gneiting & Raftery 2007 for full characterization of proper scoring rules).

**Definition 2.4.** A scoring rule $S : \Delta^{k-1} \times \{1, \ldots, k\} \to \mathbb{R} \cup \{\pm\infty\}$ is called proper if

$$\arg\min_p \mathbb{E}_{Y \sim q}[S(p, Y)] = q$$

It is easy to verify that both NLL and Brier score are proper scoring rules. Interestingly, as shown by Bröcker (2009) any proper scoring rule can be decomposed into two components: The first, refinement, quantifies the variance within $Y$ given a model's prediction, essentially capturing the model's precision. The second, calibration/reliability, assesses how well the model's predictions align with the actual outcomes.

# 3. Post-Hoc Calibration Methods

Post-hoc calibration is typically performed using either a hold-out calibration set or the validation set. Let $\mathcal{V} = \{(x_i, y_i)\}_{i=1}^m$ represent an independent calibration set (i.i.d) separate from the training set $\mathcal{T}$. For clarity, let $\hat{g}$ denote the fitted post-hoc model, trained on calibration set $\mathcal{V}_{\hat{p}} = \{(\hat{p}(x_i), y_i) \mid (x_i, y_i) \in \mathcal{V}\}$ We will refer to $\hat{g}$ as the **calibration map**.

## 3.1. Parametric Methods

Platt (1999) introduced a logistic-regression-based calibration for binary predictions, motivated by the empirical fit of a sigmoid to SVM outputs. Later, Kull et al. (2017) demonstrated that the logistic relation assumption holds in the binary case when the distributions of model outputs conditioned on the outcome are Gaussian. This insight led to their suggested binary calibrator extension named Beta calibration modeling conditional distribution using Beta distribution. For multi-class neural network settings, Guo et al. (2017) proposed three natural extensions: Matrix Scaling (**MS**), Vector Scaling (**VS**) and Temperature Scaling (**TS**). MS essentially applies multinomial logistic regression to the logits $z(x_i)$ input prior to the final softamx layer $\sigma_{SM}$ while VS is the specific case where the transformation matrix is constrained to be diagonal. TS simplifies the calibration approach further by enforcing a uniform scaling factor $T$ across all logits and is fitted by solving the following NLL minimization optimization problem:

$$\hat{T} = \arg\min_T \sum_{i=1}^m \sum_{l=1}^k -(y_i)_l \cdot log(\sigma_{SM}(\frac{z(x_i)}{T})_l) \quad (1)$$

The corresponding calibration map is then given by $\hat{g}(z(x_i)) = \sigma_{SM}(\frac{z(x_i)}{\hat{T}})$. Despite its simplicity, TS has consistently demonstrated strong performance on conf-ECE and remains a preferred method for post-hoc calibration. It has further inspired several variations, including Mozafari et al. 2018; Ji et al. 2019; Ding et al. 2021. Kull et al. (2019) proposed Dirichlet calibration as a natural extension to Beta calibration, noting slight differences from MS.

## 3.2. Non-Parametric Methods

The two prevalent non-parametric methods are **HB** and **IR**. HB introduced by Zadrozny (2001) for binary classification. It divides $\mathcal{V}_{\hat{p}}$ into $B$ pre-determined quantiles based on $\hat{p}(x_i)$ values, the estimate for each bin is computed as the observed proportion in the calibration set. The method is considered non-parametric since it makes no distributional assumptions. Naeini et al. (2015) suggested a further improvement to HB by employing a Bayesian approach to aggregate multiple binning choices. IR, proposed by (Zadrozny & Elkan, 2002) fits a calibration map $\hat{g}$ under the sole assumption of monotonicity. Formally, IR is characterized as

$$\hat{g} = \arg\min_g \sum_{i=1}^m \big(g(\hat{p}(x_i)) - y_i\big)^2 \quad (2)$$
$$\text{s. t } g(\hat{p}(x_i)) \leq g(\hat{p}(x_j)) \text{ when } x_i \preceq x_j$$

In the binary 1-D case, the problem is solved by first ordering the data and then applying PAVA (Pool Adjacent Violators Algorithm Ayer et al. 1955) fitting a piecewise constant function to the calibration map with no further assumptions or parameters to specify. Moreover, it can be shown that the fitted $\hat{g}$ is the solution for any proper scoring rule, proof and further details are supplied in Appendix D.

## 3.3. Non-Parametric Multi-Class Extensions and Assumptions

**OvR**. Both HB and IR can be extended to multi-class settings by decomposing the problem into $k$ one-vs-rest binary subproblems and normalizing the prediction scores by dividing them by their sum, we refer to these methods as IR-OvR and HB-OvR respectively. This induction is justified when one assumes **Category Independence**, formally expressed as $P(Y_i \mid \hat{p}(X)) = P(Y_i \mid \hat{p}(X)_i)$.

**sCW**. To address inefficiencies arising from small class priors or datasets, Patel et al. (2020) introduce Shared Class-Wise (sCW) training, which merges binary sub-problems for classes with similar priors. When all sub-problems are trained jointly we name the underlying assumption **Permutation Invariance** formally assuming $P(Y_i \mid \hat{p}(X)) = P(Y_{\sigma(i)} \mid \sigma(\hat{p}(X)))$ where $\sigma$ is a permutation defined such that $\sigma(\hat{p}(X))_j = \hat{p}(X)_{\sigma^{-1}(j)}$. Applying Permutation Invariance to a calibration set in our settings involves flattening all the predictions, thus, we term Flattened Isotonic Regression (**FIR**) to be the result of applying sCW training to all classes together:

$$\tilde{g}_{FIR} = \arg\min_g -\sum_{i=1}^m \sum_{l=1}^k (y_i)_l \cdot log\big(g(\hat{p}(x_i)_l)\big) \quad (3)$$
$$+ \big(1 - (y_i)_l\big) \cdot log(1 - g(\hat{p}(x_i)_l))$$

for monotonic function $g$. Patel et al. (2020) previously

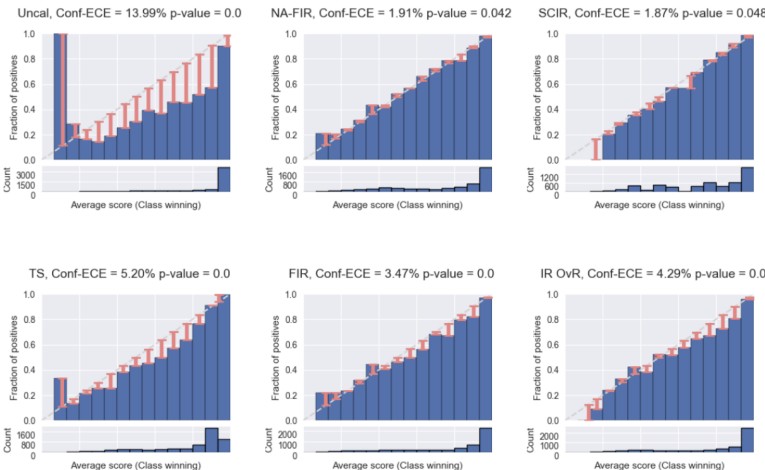

*Figure 1.* Confidence Reliability plots for NG20 BERT-Large-Uncased classifier, p-value is calcluated as suggested by Vaicenavicius et al. (2019) under the null of perfectly calibrated procedure. As can be seen the uncalibrated model is highly over-confident and both our suggested methods are the only ones where we get positive p-value.

reported FIR without normalization, whereas Zhang et al. (2020) has applied the straightforward correction for Category Independence assumption:

$$\hat{g}_{FIR}(p) = \left( \frac{\tilde{g}_{FIR}(p_1)}{\sum_{l=1}^{k} \tilde{g}_{FIR}(p_l)}, \ldots, \frac{\tilde{g}_{FIR}(p_k)}{\sum_{l=1}^{k} \tilde{g}_{FIR}(p_l)} \right)$$

Notably, FIR is **Order Preserving**, meaning $p_i \geq p_j \to \hat{g}(p)_i \geq \hat{g}(p)_j$ for all $p \in \Delta^{k-1}$. These types of structural assumptions have also been outlined in the context of parametric models (Rahimi et al., 2020).

Alternatively Gupta & Ramdas (2022) proposed a M2B framework, essentially suggesting to align calibration method with calibration error objection. For example, when confidence calibrated (Definition 2.3) is of interest, HB should be trained using a calibration set containing only the top predicted probability and a $\{0, 1\}$ encoding to indicate whether the top predicted class matches the actual class. Notably, they report SOTA performance without normalizing the output predictions.

### 3.4. Other Related Work

It is worth noting that various calibration approaches have been proposed, including post hoc methods (see Silva Filho et al. 2023 for a more comprehensive discussion), regularization techniques, implicit calibration strategies, and uncertainty estimation methods. For a comprehensive review of these methods in the context of deep learning, see Abdar et al. (2021); Gawlikowski et al. (2023).

## 4. Normalization Aware IR Methods

**Motivation.** Examining the previously proposed non-parametric multi-class extensions and their respective as-

*Table 1.* Multi-Class extensions and their underlying assumptions

| Calibration Method | Compared Methods | | | | | Our Methods | |
|---|---|---|---|---|---|---|---|
| | MS | VS | TS | IR OvR | FIR | SCIR | NA-FIR |
| Order Preserving | × | × | ✓ | × | ✓ | × | ✓ |
| Permutation Invariant | × | × | ✓ | × | ✓ | ✓ | ✓ |
| Relaxing Category Independence | ✓ | ✓ | ✓ | × | × | ✓ | ✓ |

sumptions reveals the key weakness we want to address - they all are trained without taking into consideration their actual normalized outcomes.

We believe that providing valid probability predictions is essential for building trust among practitioners, especially in decision-making frameworks. However, most existing non-parametric calibration methods for multi-class settings operate under the assumption of category independence, where each class is calibrated separately and if predictions are normalized it is only done post calibration.

This modeling choice introduces a key limitation as it is known that such marginal (classwise) calibration is strictly weaker than full (canonical) calibration (Vaicenavicius et al., 2019). For example both prediction vectors [0.8, 0.2,0] and [0.8, 0.1, 0.1] share the same top-1 prediction but their underlying uncertainty profiles can differ substantially.

While we aim to relax the Category Independence assumption, we justify Order Preserving assumption as a desirable property. As underlying models improve, it becomes increasingly reasonable for a calibration method to preserve the ranking of predicted probabilities from the original model.

Furthermore, the traditional OvR approach suffers from high

*Table 2.* Average ranking comparison between different calibration methods across different datasets and various metrics. Percentages indicating how often a method achieved the lowest score for a given metric across the tested models. The best-performing calibrator is highlighted in bold.

| METRIC | DATASET | AVERAGE RANKING | | | | | | | |
|---|---|---|---|---|---|---|---|---|---|
| | | NA-FIR | SCIR | FIR | IR OvR | TS | VS | MS | UNCALIBRATED |
| NLL | 20NG | **1.4 (70%)** | 6.0 (0%) | 2.8 (10%) | 7.3 (0%) | 3.3 (10%) | 3.2 (10%) | 5.3 (0%) | 6.7 (0%) |
| | CIFAR10 | **1.2 (75%)** | 1.8 (25%) | 4.0 (0%) | 7.0 (0%) | 5.8 (0%) | 4.8 (0%) | 3.5 (0%) | 8.0 (0%) |
| | CIFAR100 | **1.5 (62%)** | 4.4 (15%) | 3.3 (0%) | 7.6 (0%) | 2.8 (8%) | 3.8 (8%) | 6.7 (8%) | 5.9 (0%) |
| | FOOD101 | 2.5 (0%) | 5.8 (0%) | 4.0 (0%) | 7.9 (0%) | **1.6 (62%)** | 2.9 (38%) | 7.0 (0%) | 4.4 (0%) |
| | IMAGENET | **1.0 (100%)** | 4.9 (0%) | 2.0 (0%) | 8.0 (0%) | 3.4 (0%) | 5.1 (0%) | 5.9 (0%) | 5.8 (0%) |
| | R52 | **1.7 (50%)** | 5.1 (0%) | 3.2 (10%) | 8.0 (0%) | 2.1 (30%) | 5.1 (10%) | 6.7 (0%) | 4.1 (0%) |
| | YR | 3.2 (14%) | 5.3 (5%) | 5.2 (0%) | 5.2 (0%) | 4.9 (0%) | 3.2 (5%) | **1.5 (77%)** | 7.6 (0%) |
| CONF-ECE | 20NG | **2.4 (10%)** | 2.6 (60%) | 3.1 (30%) | 3.9 (0%) | 6.9 (0%) | 4.4 (0%) | 4.9 (0%) | 7.8 (0%) |
| | CIFAR10 | 4.8 (0%) | **1.0 (100%)** | 5.8 (0%) | 5.8 (0%) | 2.8 (0%) | 4.0 (0%) | 4.0 (0%) | 8.0 (0%) |
| | CIFAR100 | 2.5 (8%) | **1.2 (85%)** | 4.7 (0%) | 5.8 (0%) | 3.1 (0%) | 4.0 (8%) | 7.4 (0%) | 7.3 (0%) |
| | FOOD101 | **2.6 (25%)** | 3.2 (25%) | 4.4 (0%) | 4.9 (12%) | 2.8 (38%) | 3.4 (0%) | 7.6 (0%) | 7.1 (0%) |
| | IMAGENET | **1.4 (75%)** | 2.0 (25%) | 3.0 (0%) | 5.5 (0%) | 4.0 (0%) | 6.1 (0%) | 7.4 (0%) | 6.6 (0%) |
| | R52 | **2.6 (20%)** | 2.9 (50%) | 3.6 (10%) | 6.3 (0%) | 4.8 (0%) | 3.8 (10%) | 5.1 (10%) | 6.9 (0%) |
| | YR | **3.0 (18%)** | 3.2 (41%) | 4.1 (9%) | 3.4 (14%) | 6.0 (0%) | 4.8 (5%) | 3.7 (14%) | 7.9 (0%) |
| BS | 20NG | **2.1 (40%)** | 4.3 (0%) | 2.8 (20%) | 2.7 (30%) | 4.9 (0%) | 4.9 (10%) | 7.0 (0%) | 7.3 (0%) |
| | CIFAR10 | 2.2 (0%) | **1.0 (100%)** | 3.2 (0%) | 3.5 (0%) | 6.2 (0%) | 5.5 (0%) | 6.2 (0%) | 8.0 (0%) |
| | CIFAR100 | **2.1 (31%)** | 2.5 (46%) | 3.6 (0%) | 5.3 (0%) | 3.8 (8%) | 4.6 (8%) | 7.4 (8%) | 6.7 (0%) |
| | FOOD101 | 3.1 (12%) | 4.6 (0%) | 3.0 (0%) | 5.6 (12%) | **2.4 (38%)** | 3.6 (38%) | 7.8 (0%) | 5.9 (0%) |
| | IMAGENET | **1.2 (75%)** | 3.4 (0%) | 2.4 (0%) | 6.5 (0%) | 4.1 (12%) | 5.8 (12%) | 6.6 (0%) | 6.0 (0%) |
| | R52 | **2.4 (50%)** | 4.9 (10%) | 2.9 (0%) | 5.7 (0%) | 3.9 (20%) | 3.9 (20%) | 6.8 (0%) | 5.5 (0%) |
| | YR | 3.8 (9%) | 6.0 (5%) | 4.8 (0%) | 3.0 (5%) | 5.7 (0%) | 3.1 (5%) | **1.8 (77%)** | 7.8 (0%) |
| CW-ECE | 20NG | 4.0 (0%) | 3.8 (10%) | 3.7 (10%) | **1.6 (60%)** | 7.0 (0%) | 5.0 (0%) | 3.1 (20%) | 7.8 (0%) |
| | CIFAR10 | 4.2 (0%) | **2.0 (50%)** | 5.5 (0%) | 2.5 (25%) | 7.0 (0%) | 4.2 (0%) | 2.5 (25%) | 8.0 (0%) |
| | CIFAR100 | 2.6 (23%) | **2.4 (46%)** | 4.0 (8%) | 4.2 (8%) | 3.7 (0%) | 4.7 (15%) | 7.3 (0%) | 7.1 (0%) |
| | FOOD101 | 3.0 (0%) | 3.2 (38%) | 3.9 (0%) | 5.6 (0%) | **2.5 (25%)** | 3.8 (38%) | 7.4 (0%) | 6.6 (0%) |
| | IMAGENET | 3.6 (0%) | 6.1 (0%) | 3.1 (12%) | 4.4 (0%) | **2.0 (50%)** | 4.1 (25%) | 5.4 (12%) | 7.2 (0%) |
| | R52 | 3.8 (0%) | 5.0 (0%) | **2.7 (40%)** | 4.5 (10%) | 4.7 (10%) | 2.8 (40%) | 7.2 (0%) | 5.3 (0%) |
| | YR | 4.0 (18%) | 4.7 (0%) | 4.7 (0%) | **2.3 (14%)** | 6.4 (0%) | 4.1 (0%) | 2.1 (68%) | 7.7 (0%) |
| TECE | 20NG | 3.4 (20%) | 3.1 (10%) | 4.5 (10%) | **2.9 (30%)** | 5.6 (0%) | 5.0 (20%) | 3.7 (10%) | 7.8 (0%) |
| | CIFAR10 | 2.8 (25%) | **1.8 (50%)** | 4.5 (0%) | 2.5 (25%) | 6.8 (0%) | 4.8 (0%) | 5.0 (0%) | 8.0 (0%) |
| | CIFAR100 | 2.2 (23%) | **1.7 (62%)** | 4.3 (0%) | 4.5 (0%) | 4.5 (0%) | 4.6 (8%) | 7.6 (0%) | 6.6 (8%) |
| | FOOD101 | **2.5 (12%)** | 2.6 (25%) | 3.0 (25%) | 5.8 (12%) | 4.0 (12%) | 4.8 (12%) | 7.6 (0%) | 5.8 (0%) |
| | IMAGENET | 2.9 (12%) | **2.1 (50%)** | 2.5 (0%) | 8.0 (0%) | 5.1 (0%) | 6.4 (0%) | 6.1 (0%) | 2.9 (38%) |
| | R52 | 6.1 (0%) | **2.7 (30%)** | 5.3 (10%) | 3.6 (10%) | 5.1 (0%) | 3.6 (10%) | 3.2 (40%) | 6.4 (0%) |
| | YR | 3.9 (18%) | 5.3 (0%) | 4.7 (5%) | 2.7 (14%) | 6.0 (0%) | 3.5 (9%) | **2.0 (55%)** | 7.7 (0%) |

variance as data is sparse in critical areas, particularly for overconfident models in scenarios with limited calibration data and many classes. We find it reasonable to assume that a model's overconfidence or under-confidence exhibits similar characteristics across different classes effectively assuming Permutation Invariance.

Thus, we suggest two alternatives that aim to leverage the empirical observations from sCW and M2B frameworks, yet are designed to incorporate normalization within the problem formulation.

## 4.1. Normalized Aware Flattened Isotonic Regression

**NA-FIR** is what we believe to be the most appropriate non-parametric analogue to TS. Since NLL is a proper scoring rule, minimization incentivize calibration improvement (see stated decomposition in Section 2.2). Thus we formulate

the problem in a straightforward manner:

**Formulation.** Denote the isotonic function class as $\mathcal{G} = \{g : [0,1] \rightarrow \mathbb{R}^+ \mid g(u) \leq g(v) \text{ if } u \leq v\}$ we define NA-FIR NLL minimization problem as follows:

$$\tilde{g}_{\text{NA-FIR}} = \arg\min_{g \in \mathcal{G}} \sum_{i=1}^{m} \sum_{l=1}^{k} -(y_i)_l \cdot \log\left(\frac{g(\hat{p}(x_i)_l)}{\sum_{j=1}^{k} g(\hat{p}(x_i)_j)}\right) \quad (4)$$

and, accordingly:

$$\hat{g}_{\text{NA-FIR}}(p) = \left(\frac{\tilde{g}_{\text{NA-FIR}}(p_1)}{\sum_{l=1}^{k} \tilde{g}_{\text{NA-FIR}}(p_l)}, \dots, \frac{\tilde{g}_{\text{NA-FIR}}(p_k)}{\sum_{l=1}^{k} \tilde{g}_{\text{NA-FIR}}(p_l)}\right) \quad (5)$$

Note that while TS is applied to the softmax function (Equation (1)), the only difference in our normalized aware isotonic formulation lies in the use of derived probabilities

instead of direct logits, which is not a fundamental change as the exponent is a monotonic function.

Both FIR and NA-FIR assume Permutation Invariance and ensure Order Preservation for every prediction point. But while FIR optimization (Equation (3)) doesn't account for the collateral effects of the fitted isotonic values, NA-FIR optimization (Equation (4)) incorporates the normalization term, thereby relaxing the limitations of the Category Independence assumption.

**Fitting Process.** It is important to highlight that this problem is no longer convex, which means that PAVA cannot be directly applied as a solution. Consequently, we explored alternative methods for searching constrained local minima. Given that PAVA when applied to the non-normalized isotonic problem, identifies the optimal bin boundaries for minimizing the NLL in the flattened binary case, it provides both a reasonable initialization point and an opportunity to reduce the effective search space. Leveraging the isotonic block structure identified by PAVA allowed us to include more thorough search techniques, as outlined in our proposed algorithm Algorithm 1. The algorithm employs Markov Chain Monte Carlo (MCMC) optimization, also known as simulated annealing, achieving stable convergence while still allowing the practitioner to specify both the minimal number of bins and the granularity of fitted values. Further details can be found in Appendix A.2.

---

**Algorithm 1** MCMC Blockwise Normalized Aware Flattened Isotonic Optimization

---

1: **Input:** Data $X, Y$,
   hyperparameters $\epsilon_{\text{change}}, \beta$, num_iterations,
   min_blocks, split_size_threshold
2: Flatten $X$ and $y$: $x_{\text{flatten}} = X.\text{flatten}(), y_{\text{flatten}} = Y.\text{flatten}()$
3: Initialize block structure and fits:
   [blocks, block_fits] $\leftarrow$ PAVA($x_{\text{flatten}}, y_{\text{flatten}}$)
4: Refine block structure: [blocks, block_fits] $\leftarrow$
   split_blocks(blocks, min_blocks, split_size_threshold)
5: **for** $i = 1$ **to** num_iterations **do**
6:    Select a random block: block $\sim \mathcal{U}(1, \text{len(blocks)})$
7:    Perturb block fit: $\delta \sim \mathcal{U}(\{-1, 1\})$, change $\leftarrow \epsilon_{\text{change}} \cdot \delta$
8:    Update suggested fit:
      suggested_block_fit $\leftarrow$ block_fits[block] + change
9:    Validate isotonicity of suggested_block_fit
10:   **if** likelihood > current_likelihood **then**
11:      Update: best_likelihood $\leftarrow$ likelihood,
         best_fit $\leftarrow$ suggested_block_fits
12:   **end if**
13:   **if** likelihood > current_likelihood **or**
      rand() < exp($\beta \cdot$ (likelihood − current_likelihood)) **then**
14:      Accept: block_fits $\leftarrow$ suggested_block_fits,
         current_likelihood $\leftarrow$ likelihood
15:   **end if**
16: **end for**
17: **Return:** best_block_fits

---

## 4.2. Sorted Cumulative Isotonic Regression

Motivated to develop a calibration method that aligns with the practitioner's decision-making process, particularly modeling binary problems that correspond directly to the ECE evaluated metric (e.g conf-ECE), we propose **SCIR**, which integrates ranking into the problem formulation.

**Formulation.** Let $(p, y) \in \mathcal{V}_{\hat{p}}$ be a calibration point, and let $\sigma$ be the order permutation such that $p_{\sigma^{-1}(1)} \geq \cdots \geq p_{\sigma^{-1}(k)}$, we define the sorted cumulative corresponding set for each prediction point as

$$cu_{\text{sorted}}(p, y) := \left\{ \left( \left( \sum_{\sigma(j) \leq r} p_j, r \right), \sum_{\sigma(j) \leq r} y_j \right) \,\middle|\, 1 \leq r \leq k - 1 \right\}$$

and for a set of points we define the aggregated set $\mathcal{CU}_{\text{agg}} := \bigcup_{(p,y) \in \mathcal{V}_{\hat{p}}} cu_{\text{sorted}}(p, y)$. This set contains tuples of cumulative top-$r$ scores and their associated cumulative label counts such that each $((q, r), \tilde{y}) \in \mathcal{CU}_{\text{agg}}$ belongs to $[0, 1] \times \{1, \ldots, k\} \times \{0, 1\}$.

Finally $\tilde{g}_{\text{SCIR}}$ is defined as the solution to the following minimization problem:

$$\arg\min_g - \sum_{\mathcal{CU}_{\text{agg}}} \tilde{y} \cdot \log\left(g(q, r)\right)$$
$$+ (1 - \tilde{y}) \cdot \log\left(1 - g(q, r)\right) \tag{6}$$
$$\text{s.t } g(q, r) \leq g(s, t) \text{ if } q \leq s \wedge r \leq t$$

Thus imposing monotonicity both in prediction sum and ranking index. To make predictions for a new input point the probabilities are first sorted, $\hat{g}_{\text{CSIR}}$ is applied, a difference is taken and then the results are transformed back into the original order. Formally the prediction is calculated as:

$$\hat{g}_{\text{SCIR}}(\hat{p}(x_i))_l = \tilde{g}_{CSIR}\left( \sum_{\sigma(j) \leq \sigma(l)} \hat{p}(x_i)_j, \sigma(l) \right)$$
$$- \tilde{g}_{CSIR}\left( \sum_{\sigma(j) \leq \sigma(l)-1} \hat{p}(x_i)_j, \sigma(l) - 1 \right)$$

where $\hat{g}_{\text{CSIR}}(\cdot, 0) = 0$, $\hat{g}_{\text{CSIR}}(\cdot, k) = 1$ by definition.

**Illustrative Example.** Let $k = 4$ and consider a pair $(p, y) = ([0.2, 0.4, 0.3, 0.1], [0, 0, 1, 0])$ The corresponding sorted cumulative set is:

$$cu_{\text{sorted}}(p, y) = \left\{ ([0.4, 1], 0), ([0.7, 2], 1), ([0.9, 3], 1) \right\}$$

To compute the calibrated probability for the first class (which appears at the third rank in the sorted order), we apply: $\hat{g}_{\text{SCIR}}(p)_1 = \tilde{g}_{\text{SCIR}}(0.9, 3) - \tilde{g}_{\text{SCIR}}(0.8, 2)$

We note that this formulation assumes Permutation Invariance and encodes mutual learning effectively allowing to

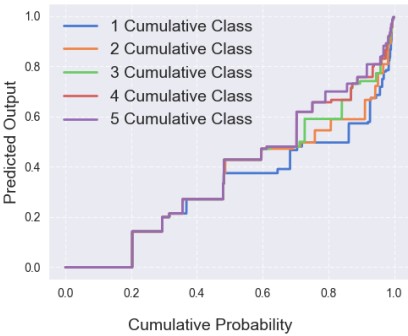

(a) BERT-large-uncased SCIR fit

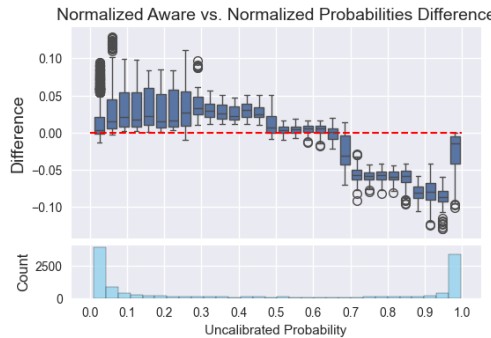

(b) BERT-large-uncased NA-FIR fits v.s FIR fits on test data

*Figure 2.* Both plots are based on the NG20 dataset with trained BERT-large-uncased classifer. The left plot illustrates fitted calibration curves for the first 5 ranks cumulative trained models, where the final prediction for each rank (cumulative class) is calculated as the difference between its cumulative value and the previous one. The right plot provides insights into the effect on test predicted probabilities comparing NA-FIR to FIR predictions as function of Uncalibrated predictions that were thresholded to lie within the $[0.01, 1]$ range and subsequently binned into 30 equal-width intervals. The lower chart depicts the corresponding bin sizes.

break Category Independence assumption, but does not guarantee Order Preservation. We justify the coordinate-wise induced order which can expand the natural isotonic induced order as we find it is reasonable to assume super-additivity of the flattened isotonic calibration map and it ensures non-negative predicted values for every cumulative class at any prediction point. Finally, in practice, we add a small $\epsilon$ to all output probabilities and divide by the sum in order to prevent zero probabilities.

**Fitting Process.** In general, solving a multi-dimensional isotonic regression is computationally demanding, with a worst-case complexity of $O(m^4 k^4)$ as demonstrated by Spouge et al. (2003) for the least square loss (Equation (2)) and by Luss & Rosset (2014) for the generalized convex case. However, we show that by utilizing one of the algorithms specified in Spouge et al. (2003) to suit this problem, the following result holds:

**Proposition 4.1.** *The cumulative sorted problem defined in Equation* (6) *can be solved in* $O(m^2 k^4)$ *worst case time using Algorithm 2.*

This solution is obtained by recursively splitting $\mathcal{CU}_{\text{agg}}$ using the surprisingly simple dynamic programming algorithm outlined in Algorithm 2. When the returned partition equals the input assigned value is the input corresponding mean. proof for Proposition 4.1 and further details are provided in Appendix A.

## 5. Experiments

### 5.1. Experiment Design

We evaluate post-hoc calibration methods on seven benchmark datasets spanning both image and text domains:

---

**Algorithm 2** 2-D Grid Sorted Cumulative Isotonic Maximal Upper Set Algorithm

1: **Input:** $\mathcal{X} = \{(k, l)_i \mid i \in [1, \dots, N]\}$, $y, w \in \mathbb{R}^n$
2: Initialize $M_{(N+1) \times k}$, $R_{N \times k}$ as zero matrices, and $b = \text{Avg}(\mathcal{X})$
3: **for** $i = N$ **to** 1 **do**
4:     **for** $j = 1$ **to** $c$ **do**
5:         **if** $j \leq l$ **then**
6:             $M[i, j] = M[i+1, j] + w_i(y_i - b); R[i, j] = j$
7:         **else if** $j > l$ **and** $(y_i - b) < 0$ **then**
8:             $M[i, j] = M[i+1, j]; R[i, j] = j$
9:         **else**
10:            $M[i, j] = M[i+1, l] + w_i(y_i - b); R[i, j] = l$
11:         **end if**
12:     **end for**
13: **end for**
14: Initialize $S = \{\}; j = c$
15: **for** $i = 1$ **to** $N$ **do**
16:     Set $(k, l) = x_i$
17:     **if** $R(i, j) \leq l$ **then**
18:         Add $i$ to $S$
19:     **end if**
20:     Update $j = R(i, j)$
21: **end for**
22: **Return:** $S$

---

CIFAR-10, CIFAR-100 (Krizhevsky, 2009), Food-101 (Bossard et al., 2014) and ImageNet-1k (Russakovsky et al., 2015) for image classification, and R52, NG20, and Yelp Review (Zhang et al., 2015) for text classification. These datasets include varying numbers of classes and validation/test set sizes, allowing us to test calibration methods across diverse scenarios (see Table 3 for further specification). Experiments were conducted using various modern DNN architectures and a few classical machine learning models. To assess the practical usefulness of calibration functions, we compared several models with different archi-

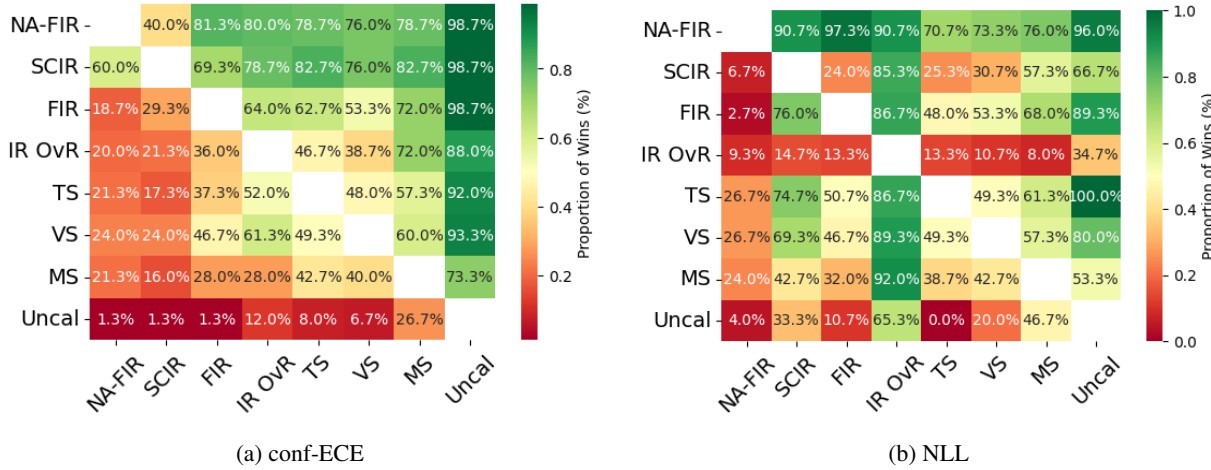

*Figure 3.* Comparison of conf-ECE and NLL between different calibration methods. Each cell in the heatmap represent the percentage of times the calibration method on the row had achieved better score then the calibration method on the column.

tectures and training regimes so overall 75 baseline models are being evaluated. For each dataset and model, we applied seven of the calibration methods outlined in Section 3 and Section 4, omitting other mentioned methods for simplicity. Detailed information on experimental setup is provided in Appendix B. The performance of these methods will be evaluated using the metrics accuracy, NLL, Brier score, conf-ECE, cw-ECE and TECE. To account for practical considerations, we also report wall-clock time. TECE is evaluated with a threshold of $1/k$, and all calibration metrics are computed using 15 bins and their respective binning schemes as defined.

### 5.2. Results

**Normalization Aware Effect.** As shown in Figure 2, while normalization awareness, as defined in Section 4.1, might appear to be a minor adjustment, it has a significant impact on predicted probabilities. Lower FIR probabilities are often inflated by NA-IR and vice versa. This phenomena effectively demonstrate that awareness has resulted in increased uncertainty in final predictions which was not accounted when fitting FIR. Looking at cumulative fits of SCIR we can notice that the first rank fit which corresponds to the suggested M2B framework is relatively "constrained" free allowing to learn conf-ECE targeted calibration map, on the other hand many overlaps and order violations seems to occur which can be problematic.

**Aggregated Comparative Results.** Table 2 demonstrates that both our proposed methods, SCIR and NA-FIR, achieve state-of-the-art performance in terms of NLL loss and conf-ECE across all datasets. In terms of conf-ECE, SCIR often achieves the best results (see Figure 3), while NA-FIR

demonstrates greater stability, consistently ranking among the top calibration methods even when it is not the best performer. It also shows that our approach for constrained NLL optimization was indeed justified, as we can see that NA-FIR not only improves NLL but also conf-ECE when compared to FIR, see Figure 1 for example. Moreover, the results clearly show that NA-FIR outperforms linear scaling methods for most datasets except for the FOOD101 dataset where TS excels and Yelp Review where there is an abundance of examples in each class. In contrast, SCIR consistently produces lower scores for NLL - an unwanted surprising property, especially when considering the well performing ECE metrics. Interestingly, this holds even when Brier score is comparable, thus we attribute this unwanted behavior to near zero probabilities arising from overlaps in the cumulative class probabilities.

A closer analysis into the gaps, as shown in Figure 7 provided in appendix C, reveals that dataset characteristics such as number of classes and data set type correlate with some of the calibration methods performance. TS performed better on image classification datasets, while both MS and IR OvR were greatly affected by the number of classes and samples provided.

With respect to wall-clock time measurements, both TS and VS are highly efficient, completing in a matter of seconds across all datasets. In contrast, SCIR and NA-FIR exhibit less favorable scaling behavior. For NA-FIR, each iteration scales linearly with the number of samples (Appendix A.2). In practice, the method required approximately 15 minutes to calibrate Yelp Review and ImageNet-1k. SCIR, by contrast, is highly sensitive to the number of classes (Proposition 4.1), taking 1 second on Yelp Review but approx-

imately 120 minutes on ImageNet-1k. All timing results were obtained on a standard single-machine setup without GPU acceleration. Nevertheless, calibration sets are often substantially smaller than training sets and the calibration overhead remains minor relative to model training time.

Lastly, re-visiting our assumptions in this empirical evaluation provides a more concrete justification for our approach. First, the superior performance of both the cumulative approach and flattening methods compared to IR OvR indicates that leveraging permutation invariance enhances the model's generalization capabilities. Second, the results highlight that order preservation is a reasonable assumption, effectively acting as a classical "variance reduction" technique, as evidenced by the more stable results observed across and within each dataset for the respective methods. Finally, the rankings and performance differences once again emphasize the effectiveness of post-hoc calibration methods in improving model reliability and robustness.

**Per Model Comparison.** We also examine how different network architectures influence post-hoc calibration. For instance, results (see Table 4 in Appendix C) for CIFAR-100, show that highly overconfident architectures (DenseNet, ResNet) benefit substantially from SCIR in comparison to other methods, while ViT and Swin being less overconfident, gain modestly yet consistently. Similarly, conf-ECE result for textual models across architectures show the same trend. Simpler models, such as FastText, Swem-Concat along with classical ML algorithms (NB, SVC), exhibit greater calibration improvements (see Figure 11 in Appendix C). In contrast, more advanced models (e.g, BERT-based) tend to show less variation across calibration methods. We can see how better in advance calibrated models still gain from post-hoc calibration but can be much less sensitive to the underlying mechanism.

## 6. Conclusions

We have proposed two non-trivial extensions for non parametric learning in post-hoc calibration that achieve state-of-the-art results. These extensions rely on simple yet effective explicit assumptions and directly incorporate them within the post-hoc optimization framework, rather than resorting to traditional k-one-vs-rest decomposition. By addressing the normalization problem, our approach demonstrates improved performance and adaptability to diverse scenarios, providing a more flexible approach for learning a calibration map. While the empirical results are encouraging, several avenues for discussion and further research remain. In particular, the relationship between calibration and refinement when analyzed through user-specific metrics like conf-ECE, warrants deeper investigation. Our results highlight a clear trade-off, emphasizing yet again the complex nature of defining a "well-calibrated" model.

## Impact Statement

This paper presents work whose goal is to advance the field of Machine Learning. There are many potential societal consequences of our work, none which we feel must be specifically highlighted here.

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

# A. Normalized Aware Methods Algorithmic Details

## A.1. SCIR

### A.1.1. PRELIMINARIES

We begin by quoting several definitions and theorems from Spouge, Wan, and Wilbur (Spouge et al., 2003), which will later be utilized to prove the proposed algorithm:

Let $X = \{x_i \mid i \in I\}$ be a partially ordered set, meaning there exists a relation $\preceq$ that is reflexive, antisymmetric, and transitive. Let $g^*$ denote the result of isotonic regression, as defined in Equation (2), for the corresponding weights $w_i \in \mathbb{R}$ and values $y_i \in \mathbb{R}$.

we denote $(g|_S)^*$ as the result of isotonic regression for $S \subseteq X$ with the respective $w_i, y_i$.

**Definition A.1.** A subset $U \subseteq X$ is called an **upper set** if $x \in U$ and $x \preceq z \implies z \in U$. A **lower set** is defined in a dual manner.

We note that upper sets are closed under arbitrary intersections and unions of upper sets. Similarly, lower sets satisfy this property. Additionally, the empty set is both an upper set and a lower set.

**Definition A.2.** A pair $(L, U)$, where $U$ is an upper set and $L = X - U$ is a lower set, is called a **projection pair** if and only if there exists a real number $d$ such that:

$$\forall x \in L, \ z \in U : (g|_L)^*(x) \le d \le (g|_U)^*(z).$$

**Theorem A.3.** *If $(L, U)$ is a projection pair, then:*

$$g^*(x) = \begin{cases} (g|_L)^*(x) & x \in L, \\ (g|_U)^*(x) & x \in U. \end{cases}$$

*Proof.* See Theorem 3.1 in (Spouge et al., 2003). □

Using Theorem A.3, we observe that by finding a projection pair, we can split the isotonic problem into sub-problems that are easier to compute. Fortunately, we have a systematic way to find such pairs.

**Definition A.4.** For $S \subseteq X$ and $S_I = \{i \mid x_i \in S\}$, define:

1. $\text{Avg}(S) = \frac{\sum_{i \in S_I} w_i \cdot y_i}{\sum_{i \in S_I} w_i}$,

2. $H_X(S) = \sum_{i \in S_I} w_i \cdot (y_i - \text{Avg}(X))$,

3. Let $U_X$ denote the set of all upper sets in $X$. A **maximal upper set** $U$ is an upper set that satisfies $U = \arg\max_{U' \in U_X} H_X(U')$. A **minimal lower set** is defined dually.

**Theorem A.5.** *Every non-empty pair $(L, U)$, where $U$ is a maximal upper set, is a projection pair. If the empty set is a maximal upper set in $X$, then $g^* \equiv H_X(X)$ for all $x_i \in X$.*

*Proof.* See Theorem 3.2, Corollary 3.1, and Corollary 3.2 in (Spouge et al., 2003). □

The result above provides a powerful framework for solving the isotonic regression problem. Specifically, the isotonic regression problem can be addressed iteratively by:

1. Identifying a maximal upper set.

2. Splitting $X$ into projection pairs.

3. Repeating the process until the maximal upper set is empty.

4. Unifying the fitted results into an isotonic function on $X$.

A.1.2. 2-D GRID DYNAMIC PROGRAMMING

We focus on $\mathcal{X} = \{x_i \mid (x_i, y_i) \in \mathcal{CU}_{\text{agg}}\}$ and the coordinate-wise partial order relation $(p, i) \preceq (q, j)$ if $p \leq q$ and $i \leq j$. Each $x \in [0, 1] \times \{1, \ldots, k-1\}$. Since we are working with isotonic functions, we can w.l.o.g stable sort $\mathcal{X}$ by:

- The first argument (cumulative probability),

- Then the second argument (cumulative class count).

This sorting allows us to treat $\mathcal{X}$ as a grid:

$$x \in \{1, \ldots, N\} \times \{1, \ldots, k-1\},$$

where all points have integer coordinates corresponding to their order index and the cumulative number of classes used. Note that $N = m \cdot (k-1)$, where $m$ is the size of the calibration set and $k$ is the number of classes.

**Definitions and Recursive Setup**

**Definition A.6.** For every grid point $(i, j) \in \mathbb{N} \times \mathbb{N}$, we define an upper set function $u : \mathbb{N} \times \mathbb{N} \to \mathcal{X}$ as:

$$u(i, j) = \{(k, l) \in \mathcal{X} \mid (i, j) \preceq (k, l)\}.$$

Note that $u(i, j)$ is an upper set for any $i, j$.

**Definition A.7.** We also define a set of upper sets for a given point on the grid as:

$$\tilde{u}(i, j) = \{U \mid u(i, j) \subseteq U, \forall (k, l) \in U : k \geq i, U \text{ is an upper set}\}.$$

**Lemma A.8.** *Let $U_{\mathcal{X}_i}$ be the set of all upper sets in $\mathcal{X}_i = \{x_k \mid k \geq i\}$. Then:*

$$\tilde{u}(i, c) = U_{\mathcal{X}_i}, \quad \text{and in particular, } \tilde{u}(1, c) = U_{\mathcal{X}}.$$

*Proof.* Let $U \in U_{\mathcal{X}_i}$. We will show that $U \in \tilde{u}(i, c)$: First, note that for every $i$, $u(i, c) = \emptyset$, as the second coordinate $c$ is, by construction, not smaller than any point in $\mathcal{X}$. Thus, $u(i, c) \subseteq U$ trivially. Second, for all $(k, l) \in \mathcal{X}_i$, $k \geq i$ by construction. This also applies to $U \subseteq \mathcal{X}_i$. Lastly, $U$ is also an upper set in $\mathcal{X}$. For any point $x \in \mathcal{X} - \mathcal{X}_i$, the first coordinate is smaller than $i$, so $U \in \tilde{u}(i, c)$, implying $U_{\mathcal{X}_i} \subseteq \tilde{u}(i, c)$.

Equality holds since $U \in \tilde{u}(i, c)$ implies $U$ is an upper set with $U \subseteq \mathcal{X}_i$. $\square$

**Lemma A.9.** *Let $x_i = (i, l)$ and $m \in \mathbb{N}$. Then:*

$$\tilde{u}(i, l+m) = \tilde{u}(i+1, l+m) \uplus \tilde{u}(i, l).$$

*Proof.* Define:

$$\tilde{u}(i, l+m) = u_{-x_i} \uplus u_{x_i},$$

where:

- $u_{-x_i} = \{U \mid U \in \tilde{u}(i, l+m), x_i \notin U\}$,

- $u_{x_i} = \{U \mid U \in \tilde{u}(i, l+m), x_i \in U\}$.

1. $u_{-x_i} = \tilde{u}(i, l+m)$: let $U \in u_{-x_i}$, If $x_i \notin U$, then $\forall (k, l) \in U : k > i$, as there is only one point in $\mathcal{X}$ for each coordinate, $u(i, l+m) = u(i+1, l+m)$ thus $U \in \tilde{u}(i+1, l+m)$, leading to $u_{-x_i} \subseteq \tilde{u}(i+1, l+m)$. The reverse follows by the same arguments, implying $u_{-x_i} = \tilde{u}(i+1, l+m)$.

2. $u_{x_i} = \tilde{u}(i, l)$: let $U \in u_{x_i}$ since $x_i \in U$ and $U$ is an upper set, then $u(i, l) \subseteq U$, meaning $U \in \tilde{u}(i, l)$. Hence, $u_{x_i} = \tilde{u}(i, l)$. the opposite direction follows as if $U' \in \tilde{u}(i, l) \to x_i \in U'$ and the two other conditions are identical for $\tilde{u}(i, l+m)$ leading to $\tilde{u}(i, l) \subseteq u_{x_i}$

Combining these results, we get:

$$\tilde{u}(i, l+m) = \tilde{u}(i+1, l+m) \uplus \tilde{u}(i, l).$$

$\square$

**Definition A.10.** Define:
$$M(i,j) = \max_{U \in \tilde{u}(i,j)} H_{\mathcal{X}}(U),$$
where $H_{\mathcal{X}}(U)$ evaluates the cumulative contribution of an upper set $U$.

**Lemma A.11.** $M(i,j)$ *can be computed recursively as:*

$$M(i,j) = \begin{cases} M(i+1,j) + v & \text{if } j \leq l, \\ \max(M(i+1,j), M(i,l)) & \text{otherwise,} \end{cases}$$

*where $x_i = (i,l)$ and $v = w_i(y_i - Avg(\mathcal{X}))$.*

*Proof.* If $j \leq l$, then $(i,j) \preceq (i,l) \implies x_i \in u(i,j)$. Since there is only one point in $\mathcal{X}$ for each coordinate, we have $u(i,j) = u(i,l)$, and:
$$u(i,l) = u(i+1,l) \cup \{(i,l)\}.$$

Thus:
$$\tilde{u}(i,j) = \{(i,l)\} \cup \{U \mid u(i+1,l) \subseteq U, \forall (k,l) \in U : \; k \geq i, U \text{ is an upper set}\}.$$

Since there is only one point in $\mathcal{X}$ already included, this reduces to:
$$\tilde{u}(i,j) = \{(i,l)\} \cup \tilde{u}(i+1,l).$$

Therefore:
$$M(i,j) = \max_{U \in \tilde{u}(i,j)} H_{\mathcal{X}}(U) = w_i(y_i - \text{Avg}(\mathcal{X})) + M(i+1,j).$$

If $j > l$, the result follows directly from Lemma 12:
$$M(i,j) = \max(M(i+1,j), M(i,l)).$$

$\square$

**Complexity Analysis**

The algorithm iterates over rows in decreasing order and columns in increasing order. For each cell $(i,j)$, the computation involves constant time operations. This leads to a complexity of $O(N \cdot k) \approx O(n \cdot k^2)$ for building $M$ and $R$. Since the algorithm recursively splits $\mathcal{X}$ into two non-empty sets, the overall complexity is $O(N^2) \approx O(n^2 k^4)$ (see Lemma 4.1 in (Spouge et al., 2003) for details).

**Prediction on the Grid**

After computing $g^*$ using the outlined method, we extend the framework to include fast predictions for new points. A straightforward generalization from the univariate case is:

$$g^*(p,k) = \begin{cases} \min\{g^*(q,l) \mid (q,l) \in \mathcal{X}\} & \text{if } (p,k) \preceq (q,l) \; \forall (q,l) \in \mathcal{X}, \\ \max\{g^*(q,l) \mid (q,l) \in \mathcal{X}, \; (q,l) \preceq (p,k)\} & \text{otherwise.} \end{cases}$$

To ensure efficiency on the grid, we define a recursive function $S$ that pre-computes the maximal prediction for any grid point $(i,j)$. Initialize:
$$S(0, \cdot) = \min\{g^*(q,l) \mid (q,l) \in \mathcal{CU}_{\text{agg}}\},$$
and for $S(i,j)$ with $x_i = (i,l)$, use the recursive formula:

$$S(i,j) = \begin{cases} S(i-1,j) & \text{if } j < l, \\ \max(g^*(x_i), S(i-1,j)) & \text{if } j \geq l. \end{cases}$$

This computation requires $O(N \cdot k)$ time for pre-computing $S(i,j)$ and $O(\log(N \cdot k))$ for a new prediction using binary search.

---

**Algorithm 3** 2-D Grid Sorted Cumulative Isotonic Maximal Upper Set Algorithm

---

1: **Input:** $\mathcal{X} = \{(k,l)_i \mid i \in [1, \ldots, N]\}$, $y, w \in \mathbb{R}^n$
2: Initialize $M_{(N+1) \times c}$, $R_{N \times c}$ as zero matrices, and $b = \text{Avg}(\mathcal{X})$
3: **for** $i = N$ **to** 1 **do**
4:     **for** $j = 1$ **to** $c$ **do**
5:         **if** $j \leq l$ **then**
6:             $M[i,j] = M[i+1,j] + w_i(y_i - b); R[i,j] = j$
7:         **else if** $j > l$ **and** $(y_i - b) < 0$ **then**
8:             $M[i,j] = M[i+1,j]; R[i,j] = j$
9:         **else**
10:            $M[i,j] = M[i+1,l] + w_i(y_i - b); R[i,j] = l$
11:         **end if**
12:     **end for**
13: **end for**
14: Initialize $S = \{\}; j = c$
15: **for** $i = 1$ **to** $N$ **do**
16:     Set $(k,l) = x_i$
17:     **if** $R(i,j) \leq l$ **then**
18:         Add $i$ to $S$
19:     **end if**
20:     Update $j = R(i,j)$
21: **end for**
22: **Return:** $S$

---

### A.2. NA-FIR

We will note that the FIR as described cannot be solved using previous publications on isotonic solutions, we thus considered alternatives for local minima searching:

1. MCMC optimization (also known as simulated annealing), that works by probabilistic search in the isotonic region.

2. Simple implementation of GD that works by calculating in each step $\frac{\partial f}{\partial g}|_{g(x)}$ and after the update projects onto the isotonic region by applying PAVA.

3. Quadratic programming based methods- SLQSP, trust-constr from scipy were considered.

4. Use a penalty approach-code violation into objective function.

where we implemented split blocks to be a partition of the current block structure to equal mass points within each block such that they are not less then $block\_size\_split\_threshold$. since largest bins are for small probabilities, to complete the bin partition smaller bins are used for the remainder. see illustrative example in Figure 4.

**Scale.** In order to compute Equation (4) we need to calculate the likelihood in each iteration which can result in complexity of $O(m \cdot k \cdot \text{num\_iterations})$. This can lead to high number of operations for high dimensional datasets. given $B$ blocks that partition the $[0,1]$ domain to non-overlapping bins $\text{Bin}_1, \ldots, \text{Bin}_B$ define:

$$b_j := |\{(i,l) \in [m] \times [k] \mid \hat{p}(x_i)_l \in \text{Bin}_j \ \wedge (y_i)_l = 1\}|$$

for $j$ in [B] and let $C \in \mathbb{N}^{m \times B}$ be a matrix where

$$c_{i,j} := |\{l \in [k] \mid \hat{p}(x_i)_l \in \text{Bin}_j\}|$$

Let $g_j$ denote the value assigned for $\text{Bin}_j$ and define $T = C \begin{bmatrix} g_1 \\ \vdots \\ g_B \end{bmatrix} \in \mathbb{R}^m$ We can rewrite the log-likelihood from Equation (4)

---

**Algorithm 4** MCMC Blockwise Normalized Aware Flattened Isotonic Optimization

---

 1: **Input:** Data $X, w, y$, hyperparameters $\epsilon_{\text{change}}, \beta$, num_iterations, min_blocks, block_split_size_threshold

 2: Flatten $X$ and $y$: $x_{\text{flatten}} = X.\text{flatten}(), y_{\text{flatten}} = y.\text{flatten}()$

 3: Initialize block structure and fits: $[\text{block\_structure}, \text{block\_fits}] \leftarrow \text{PAVA}(X, y, w)$

 4: Refine block structure: $[\text{block\_structure}, \text{block\_fits}] \leftarrow \text{split\_blocks}(\text{block\_structure}, \text{min\_blocks}, \text{block\_split\_size\_threshold})$

 5: **for** $i = 1$ **to** num_iterations **do**

 6:     Select a random block: block $\sim \mathcal{U}(1, \text{len}(\text{block\_structure}))$

 7:     Perturb block fit: $\delta \sim \{-1, 1\}$, change $\leftarrow \epsilon_{\text{change}} \cdot \delta$

 8:     Update suggested fit: suggested_block_fit $\leftarrow$ block_fits[block] + change

 9:     Validate isotonicity on suggested_block_fit

10:     **if** likelihood > current_likelihood **then**

11:         Update: best_likelihood $\leftarrow$ likelihood, best_fit $\leftarrow$ suggested_block_fits

12:     **end if**

13:     **if** likelihood > current_likelihood **or** rand() $< \exp(\beta \cdot (\text{likelihood} - \text{current\_likelihood}))$ **then**

14:         Accept: block_fits $\leftarrow$ suggested_block_fits

15:     **end if**

16: **end for**

17: **Return:** best_block_fits

---

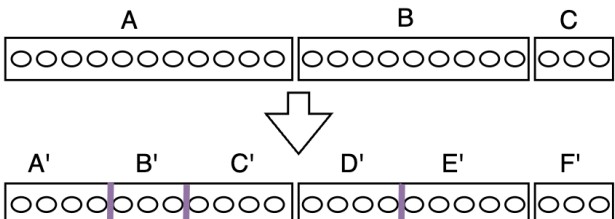

*Figure 4.* A simple example for block split with values $block\_size\_split\_threshold = 2$, $min\_blocks = 6$ and original PAVA solution of block-structure.

as follows:

$$L(g) = \sum_{i=1}^{m} \sum_{l=1}^{k} (y_i)_l \cdot \log\left(g(\hat{p}(x_i)_l)\right) - \sum_{i=1}^{m} \log\left(\sum_{j=1}^{k} g(\hat{p}(x_i)_j)\right)$$

$$= \sum_{i=1}^{B} b_j \log(g_j) - \sum_{i=1}^{m} \log(T_i)$$

Since each iteration in our optimization changes only one block value $g_j$, we can exploit this sparsity to efficiently compute the updated objective. We incrementally update the relevant entries via $T_{new} = T + C_j \cdot \epsilon_{\text{change}} \cdot \delta$, where $C_j$ is the $j$'th column and $\epsilon_{\text{change}}, \delta$ are defined in Algorithm 1. This allows us to calculate the likelihood in $O(m)$ operations per iteration.

In our experiments, we ran the MCMC algorithm with $beta = 200$, a maximum of number of $10^5$ iterations, and an early stopping criterion triggered if the best likelihood did not improve for $10^4$ iterations. This configuration consistently yielded stable results and often outperformed alternatives, even when the number of bins matched the PAVA solution. For simplicity, we report results based on this setting.

# B. Additional Experimental Details

## B.1. Dataset Descriptions

1. **CIFAR-10**: Consists of 60,000 32x32 color images in 10 classes, with 6,000 images per class.

2. **CIFAR-100**: A refinement of CIFAR-10 with 100 classes, each containing 600 images per class.

3. **Food-101**: Comprises 101 food categories with a total of 101,000 images. Each class contains 250 manually reviewed test images and 750 training images.

4. **ImageNet-1k**: Contains 1.2 million training images and 50,000 validation images across 1,000 object categories. Drawn from the ImageNet Large Scale Visual Recognition Challenge (ILSVRC).

5. **R52**: Collected from the Reuters financial newswire service in 1987. Only label categories appearing in both train and test datasets are retained.

6. **NG20**: Contains approximately 18,000 newsgroup posts on 20 topics.

7. **Yelp Review**: Comprises Yelp reviews and their respective star ratings. Extracted from the Yelp Dataset Challenge 2015 data.

*Table 3.* Comparison Study Dataset Descriptions

| Dataset | Type | # Classes | # Models Used | Validation Set Size | Test Set Size |
|---|---|---|---|---|---|
| CIFAR-10 | Image Classification | 10 | 4 | 5k | 10k |
| CIFAR-100 | Image Classification | 100 | 13 | 5k / 2.5k | 10k / 7.5k |
| Food-101 | Image Classification | 101 | 8 | 6.3k | 18.6k |
| Imagenet-1k | Image Classification | 1000 | 8 | 12.5k | 37.5k |
| R52 | Text Classification | 52 | 10 | 0.6k | 1.9k |
| NG20 | Text Classification | 20 | 10 | 1.7k | 5.3k |
| Yelp Review | Text Classification | 5 | 22 | 12.5k | 37.5k |

**Text Classifiers**

Since for text classification there were not enough datasets with many classes and pre-trained classifiers, we trained models based on (Li et al., 2022) text classification review for all datasets.

**R52 and NG20:** A total of 10 models were trained. All BERT-based models were trained with an evaluation set comprising 10% of the data for 5 epochs, using a learning rate of 1.5e-05 and a batch size of 16. The best model was selected based on evaluation set accuracy. For simpler neural network models (FastText, Kim-CNN, and SWEM-Concat), the best-performing model was chosen from the following parameter grid:

- Batch size (bs): [16, 32]

- Learning rate (lr): [1e-3, 5e-3, 1e-2]

These models used GloVe 6B with 300-dimensional embeddings. For Naive Bayes (NB) and SVM linear models 5-fold cross-validation was with Parameter grids of:

- Naive Bayes (alpha): [0.01, 0.05, 0.1, 1.0, 10.0]

- SVM (C): [0.1, 1, 3, 10, 100]

Model architecture trained:

- albert-base-v2 (link)

- bert-base-uncased (link)

- bert-large-uncased (link)

- distilbert-base-uncased (link)

- fast-text

- kim-cnn

- nb (link)

- SVC(kernel="linear") (link)

- swem-concat

- xlnet-base-cased (link)

**Yelp Review (YR):** A total of 22 models were trained. The same models as above were trained on this dataset, with the following adjustments:

- Evaluation set size reduced to 1% of the data.

- Introduced checkpointing every 500 steps.

- Added early stopping with a minimum 0.5% improvement in evaluation accuracy.

Additionally, 12 pre-trained models from HuggingFace were used:

- bert-base-uncased-hf (link)

- bert-base-cased-hf (link)

- bert-base-cased-hf-1 (link)

- bert-base-cased-hf-2 (link)

- bert-base-cased-hf-3 (link)

- bert-base-cased-hf-4 (link)

- distilbert-base-uncased-hf (link)

- distilbert-base-uncased-hf-1 (link)

- gpt2-hf (link)

- roberta-hf (link)

- ZS-deberta-base (link)

- ZS-deberta-small (link)

### C.2 Image Classifiers

**CIFAR-10:** A total of 4 models were used, all provided by the Focal Calibration Library (link):

- Densenet121

- Resnet110

- Resnet50

- Wideresnet

**CIFAR-100:**   A total of 4 models were used from the Focal Calibration Library (link), and an additional 9 models from HuggingFace:

- DenseNet121 (link)
- ResNet18 (link)
- ResNet34 (link)
- ResNet50 (link)
- Swin (link)
- Swin-1 (link)
- Vgg16 (link)
- ViT (link)
- ViT-1 (link)

**Food-101:**   All 8 models were sourced from HuggingFace:

- Swin (link)
- Swin-1 (link)
- Swin-2 (link)
- Swin-3 (link)
- ViT (link)
- ViT-1 (link)
- ViT-2 (link)
- ViT-3 (link)

**Imagenet-1k:**   All 8 models were sourced from HuggingFace (Wightman, 2019):

- Beit (link)
- ConvNeXT (link)
- Swin (link)
- ResNetRS-B (link)
- DenseNet-121 (link)
- Eva (link)
- MobileNetV3 (link)
- ViT (link)

# C. Extended Results

## C.0.1. TRAINING OPTIMIZATION

**SCIR** Empirical run time estimations for NG20 dataset for both for min-cut graph partition (using Igraph package from python) and dynamic grid partition:

we can note that our proposed dynamic grid partition runs not only much faster but gains seems to be linear. this phenomena is also replicated using much bigger dataset (cifar-100 with 10K test samples) and a comparison to randomly generated data yielding the below results:

Allowing us to scale efficiently for all the datasets included.

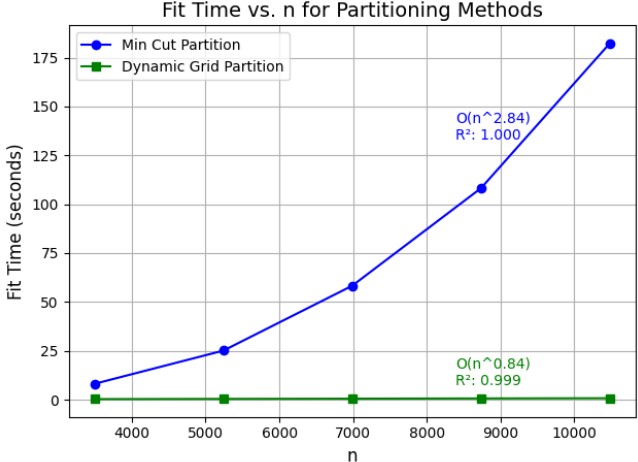

*Figure 5.* fit time in seconds for dynamic grid partition vs min cut partition

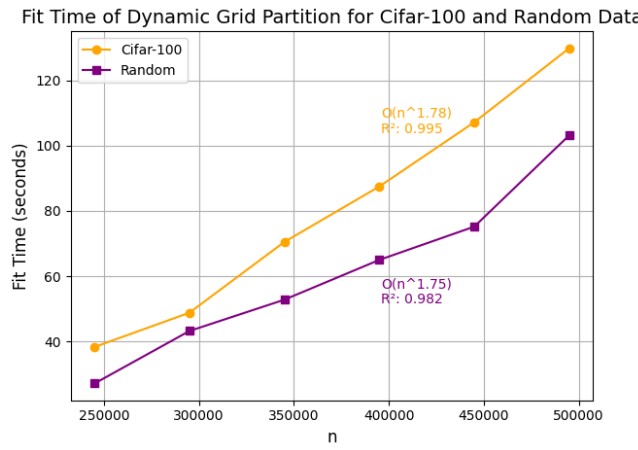

*Figure 6.* fit time in seconds both for Cifar-100 data - each time binning a few classes together to create less samples in the cumulative induce problem set and randomly generated data to account for inter-dependencies due to the structure of the problem.

### C.0.2. COMPARISONS

**Comparison of calibration methods for all metrics across datasets**

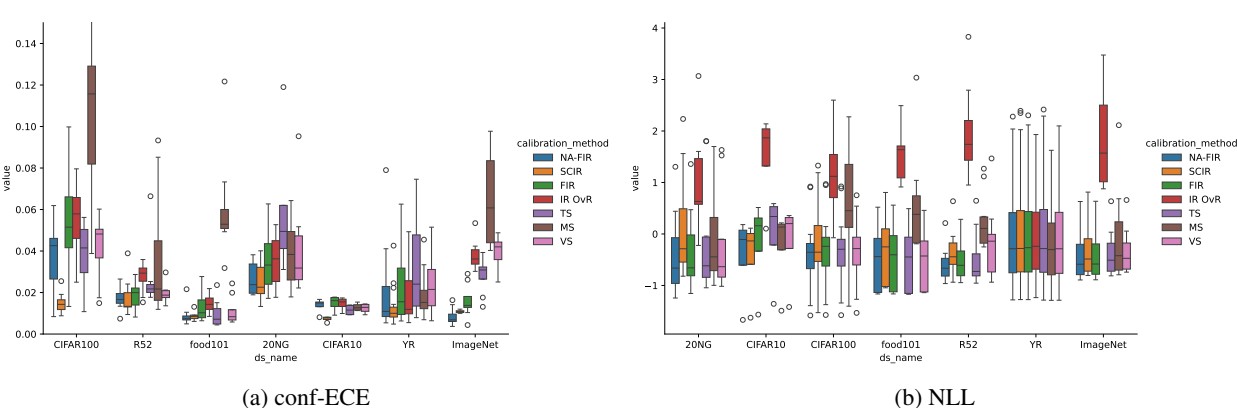

(a) conf-ECE          (b) NLL

*Figure 7.* Comparison of conf-ECE and NLL values across datasets. NLL values were normalized within each dataset by subtracting the mean and dividing by the standard deviation.

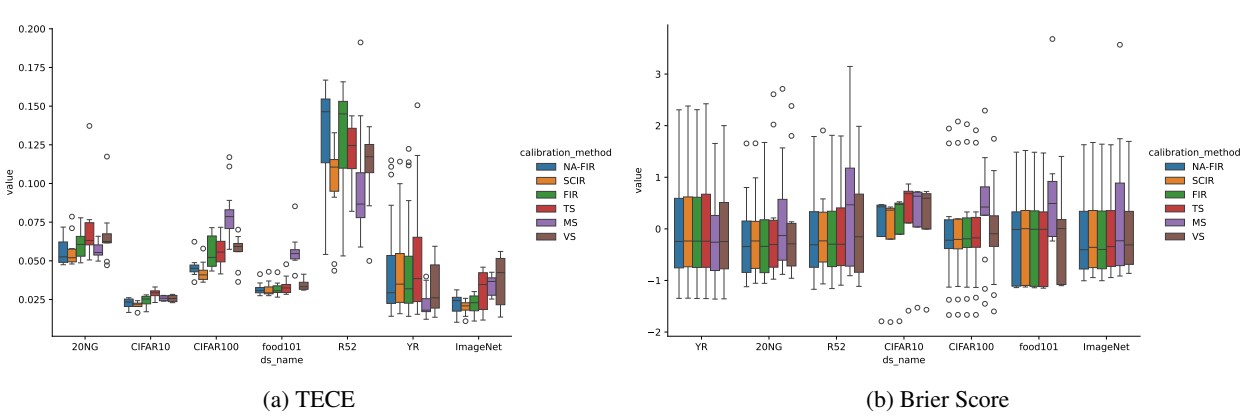

(a) TECE          (b) Brier Score

*Figure 8.* Comparison of TECE and Brier score values across datasets. Brier score values were normalized within each dataset by subtracting the mean and dividing by the standard deviation.

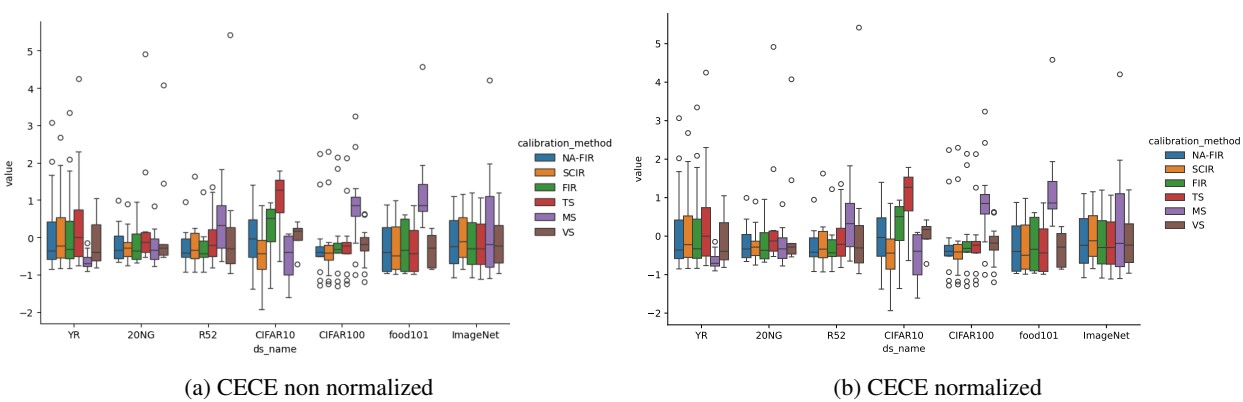

(a) CECE non normalized  (b) CECE normalized

*Figure 9.* Comparison of CECE across datasets. Both normalized and non normalized plots are presented as differences are small.

**Win Ratios for all metrics** Each cell in the heatmap represent the percentage of times the calibration method on the row had achieved better score then the calibration method on the column.

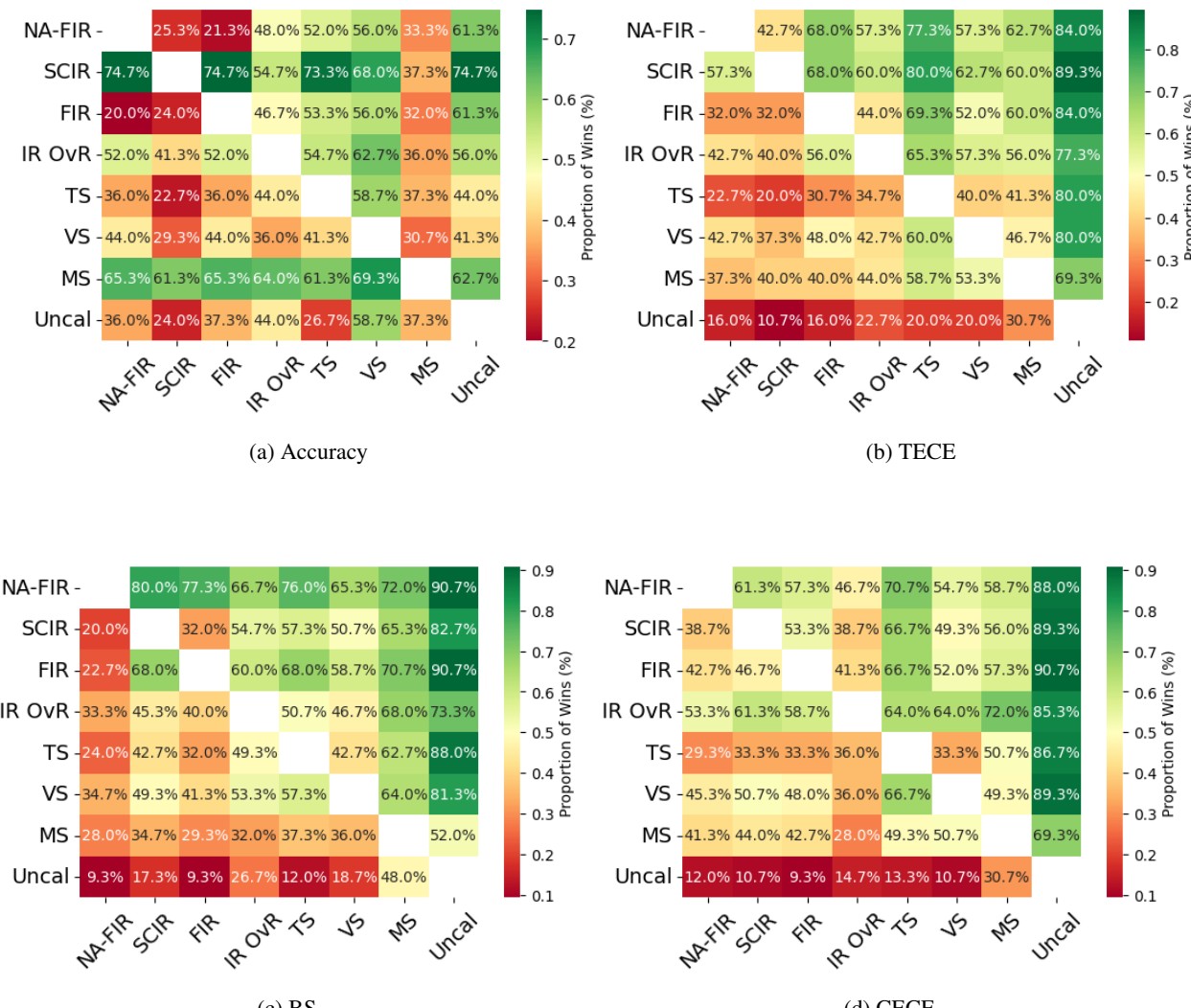

(a) Accuracy

(b) TECE

(c) BS

(d) CECE

**Model architectures results comparison** A comparison between older and newer architectures across given relevant datasets. Note that number of samples per model architecture is low.

**text models**

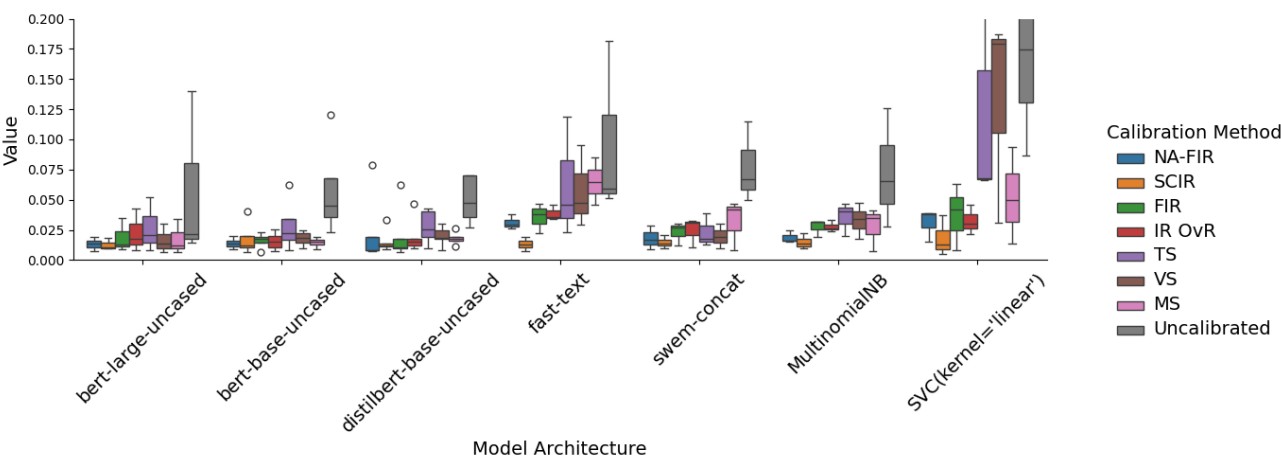

*Figure 11.* conf-ECE plotted for same model architecture and aggregated across datasets

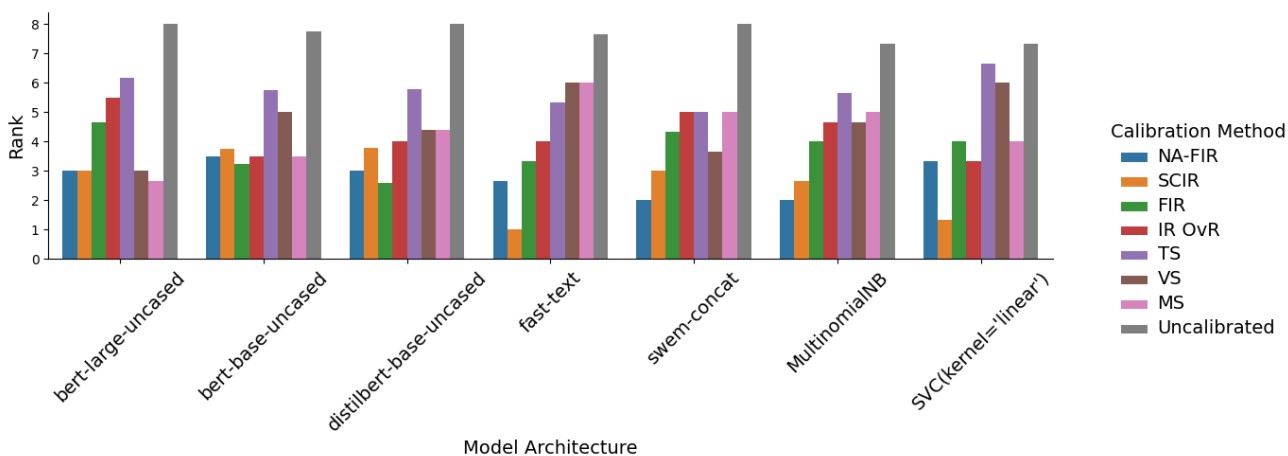

*Figure 12.* conf-ECE rank mean plotted for same model architecture and aggregated across datasets

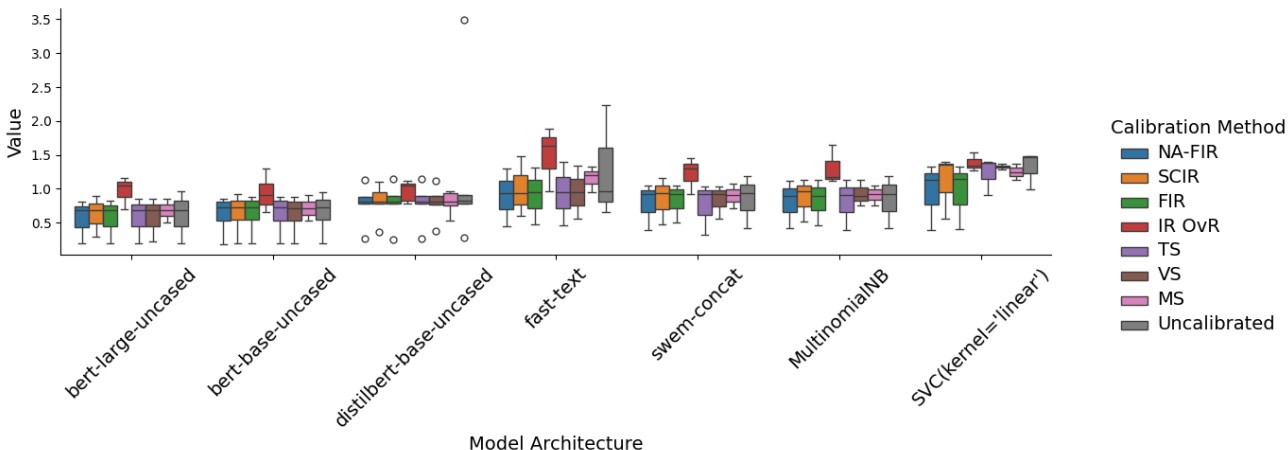

*Figure 13.* NLL plotted for same model architecture and aggregated across datasets

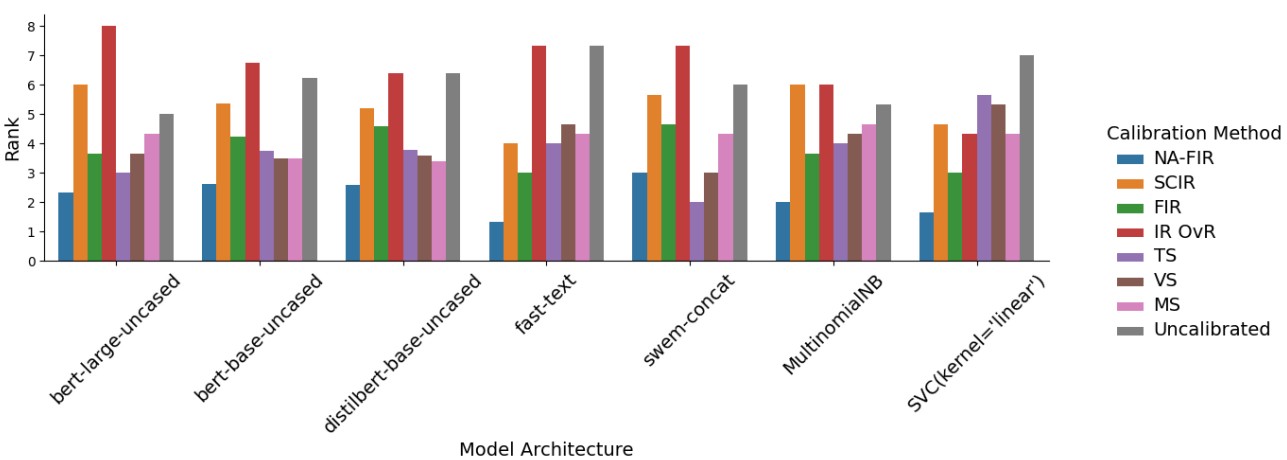

*Figure 14.* NLL rank mean plotted for same model architecture and aggregated across datasets

**Per Model tables**

*Table 4.* CIFAR 100 models comparison

| MODEL ARCHETICTURE | DENSENET121 | | | | | | RESNET110 | | | | | |
|---|---|---|---|---|---|---|---|---|---|---|---|---|
| CALIBRATION MODEL | ACCURACY | CONF-ECE | NLL | CW-ECE | BS | TECE | ACCURACY | CONF-ECE | NLL | CW-ECE | BS | TECE |
| UNCALIBRATED | 75.48% | 20.98% | 2.056 | 0.450% | 0.00446 | 17.08% | 77.27% | 19.05% | 1.792 | 0.41% | 0.004071 | 16.61% |
| NA-FIR | 75.51% | 4.91% | **1.112** | 0.229% | 0.00359 | 4.12% | 77.35% | 5.38% | **0.977** | 0.21% | 0.003297 | 4.48% |
| SCIR | 75.20% | **1.57%** | 1.204 | 0.221% | **0.00353** | **3.62%** | 77.00% | **1.27%** | 0.995 | **0.20%** | **0.003248** | **4.00%** |
| FIR | 75.51% | 7.56% | 1.194 | 0.238% | 0.00370 | 7.06% | 77.35% | 7.08% | 1.047 | 0.22% | 0.003377 | 7.14% |
| IR OVR | 75.28% | 7.96% | 1.718 | **0.218%** | 0.00370 | 5.27% | 77.11% | 7.36% | 1.539 | 0.21% | 0.003394 | 5.87% |
| TS | 75.48% | 5.06% | 1.190 | 0.236% | 0.00371 | 7.14% | 77.27% | 5.04% | 1.045 | 0.21% | 0.003383 | 7.17% |
| VS | 75.84% | 4.83% | 1.169 | 0.219% | 0.00361 | 6.07% | 77.28% | 5.36% | 1.048 | 0.22% | 0.003387 | 7.02% |
| MS | 71.79% | 13.87% | 1.860 | 0.361% | 0.00433 | 8.09% | 73.88% | 12.91% | 1.843 | 0.34% | 0.004118 | 7.95% |

Both models were used in (Mukhoti et al., 2020)

| MODEL ARCHETICTURE | SWIN | | | | | | VIT | | | | | |
|---|---|---|---|---|---|---|---|---|---|---|---|---|
| CALIBRATION MODEL | ACCURACY | CONF-ECE | NLL | CW-ECE | BS | TECE | ACCURACY | CONF-ECE | NLL | CW-ECE | BS | TECE |
| UNCALIBRATED | 87.19% | 3.28% | 0.4208 | 0.15% | 0.00185 | 5.67% | 89.63% | 6.36% | 0.448 | 0.160% | 0.001679 | 7.14% |
| NA-FIR | 87.19% | **0.84%** | **0.4116** | 0.14% | **0.00183** | **4.61%** | 89.57% | 1.68% | **0.368** | 0.121% | **0.001517** | 4.52% |
| SCIR | 87.20% | 0.88% | 0.4698 | **0.13%** | 0.00185 | 4.91% | 89.43% | **1.09%** | 0.430 | **0.118%** | 0.001525 | **4.39%** |
| FIR | 87.19% | 1.33% | 0.4121 | 0.14% | **0.00183** | 5.22% | 89.56% | 2.84% | 0.379 | 0.119% | 0.001538 | 4.80% |
| IR OVR | 86.49% | 3.30% | 1.3629 | 0.16% | 0.00197 | 6.54% | 89.55% | 3.86% | 1.234 | 0.134% | 0.001583 | 5.00% |
| TS | 87.19% | 1.08% | 0.4112 | 0.14% | 0.00183 | 4.71% | 89.63% | 2.96% | 0.396 | 0.122% | 0.001571 | 5.89% |
| VS | 86.89% | 1.75% | 0.4347 | 0.16% | 0.00190 | 5.59% | 89.59% | 3.67% | 0.439 | 0.137% | 0.001634 | 6.57% |
| MS | 83.08% | 7.10% | 0.6514 | 0.22% | 0.00252 | 7.86% | 88.75% | 5.36% | 0.537 | 0.163% | 0.001790 | 6.62% |

Both models are publicly available on HuggingFace.

*Table 5.* BERT and XLNet calibration comparison

| Model Archetecture | BERT-Base-Uncased | | | | | | XLNet-Base-Cased | | | | | |
|---|---|---|---|---|---|---|---|---|---|---|---|---|
| Calibration model | Accuracy | conf-ECE | NLL | cw-ECE | BS | TECE | Accuracy | conf-ECE | NLL | cw-ECE | BS | TECE |
| Uncalibrated | 95.90% | 2.34% | 0.1927 | 0.169% | 0.001341 | 13.75% | 94.40% | 2.38% | 0.2626 | 0.200% | 0.001654 | 13.40% |
| NA-FIR | 95.80% | 1.46% | **0.1873** | 0.163% | **0.001272** | 14.54% | 94.55% | 1.77% | **0.2513** | 0.206% | **0.001627** | 15.88% |
| SCIR | 95.90% | **1.34%** | 0.1992 | 0.162% | 0.001324 | 10.90% | 94.29% | 2.41% | 0.3746 | 0.198% | 0.001785 | 13.28% |
| FIR | 95.90% | 1.75% | 0.1987 | 0.163% | 0.001279 | 14.20% | 94.55% | 2.21% | 0.2562 | 0.204% | 0.001649 | 15.55% |
| IR OvR | 94.61% | 2.51% | 1.2967 | 0.189% | 0.001583 | **7.65%** | 94.29% | 2.65% | 1.2616 | 0.210% | 0.001782 | 10.82% |
| TS | 95.90% | 2.00% | 0.1909 | 0.175% | 0.001339 | 13.62% | 94.40% | 2.34% | 0.2551 | 0.206% | 0.001644 | 12.51% |
| VS | 95.95% | 1.59% | 0.1995 | **0.160%** | 0.001304 | 12.56% | 94.81% | 2.12% | 0.2588 | **0.197%** | 0.001634 | 13.58% |
| MS | 94.71% | 1.47% | 0.5322 | 0.194% | 0.001537 | 7.92% | 93.41% | **1.52%** | 0.5712 | 0.232% | 0.001887 | **8.05%** |

R52 models - BERT base and XLNet base cased , both models were trained for this thesis.

| Model Archetecture | BERT-Base-Uncased | | | | | | XLNet-Base-Cased | | | | | |
|---|---|---|---|---|---|---|---|---|---|---|---|---|
| Calibration model | Accuracy | conf-ECE | NLL | cw-ECE | BS | TECE | Accuracy | conf-ECE | NLL | cw-ECE | BS | TECE |
| Uncalibrated | 75.13% | 12.01% | 0.9437 | 1.309% | 0.01886 | 11.00% | 75.36% | 12.80% | 0.9561 | 1.386% | 0.01881 | 10.83% |
| NA-FIR | 75.17% | 2.02% | **0.8583** | 0.720% | 0.01733 | **4.87%** | 75.25% | **1.89%** | 0.8468 | 0.687% | **0.01716** | 4.99% |
| SCIR | 75.11% | 4.01% | 0.9158 | 0.805% | 0.01756 | 4.93% | 74.81% | 2.15% | 0.9400 | **0.652%** | 0.01748 | **4.96%** |
| FIR | 75.17% | **1.76%** | 0.8743 | 0.712% | **0.01731** | 4.88% | 75.27% | 3.47% | 0.8914 | 0.680% | 0.01731 | 5.56% |
| IR OvR | 75.19% | 1.86% | 1.0050 | 0.695% | 0.01751 | 5.46% | 75.34% | 4.19% | 1.1708 | 0.692% | 0.01745 | 5.59% |
| TS | 75.13% | 6.20% | 0.8844 | 0.852% | 0.01769 | 7.64% | 75.36% | 4.69% | **0.8467** | 0.735% | 0.01739 | 6.50% |
| VS | 75.23% | 2.43% | 0.8847 | 0.814% | 0.01775 | 6.80% | 75.13% | 3.35% | 0.8536 | 0.731% | 0.01751 | 6.20% |
| MS | 74.98% | 1.95% | 0.9063 | **0.643%** | 0.01808 | 5.55% | 74.45% | 4.88% | 0.9021 | 0.802% | 0.01807 | 6.44% |

NG20 models - BERT base and XLNet base cased , both models were trained for this thesis.

## D. Isotonic Regression Properties

**Theorem D.1.** *Let $S : [0,1] \times \{0,1\} \to (-\infty, \infty)$ be a proper scoring rule and $\mathcal{D} = \{(z_1, y_1), ..., (z_n, y_n)\}$. The isotonic optimization problem that is induced by this scoring rule and total order $\leq$:*

$$\min_g \sum_{i=1}^n S(g(z_i), y_i) \cdot w_i \quad s.t. \quad g(z_i) \leq g(z_j) \text{ when } z_i \leq z_j$$

*is solved by:*

$$\hat{y}_i = \min_l \max_u \frac{\sum_{j=u}^l w_j y_j}{\sum_{j=u}^l w_j} \quad s.t. \quad u \leq i \leq l$$

Before presenting the proof, it is worth noting that Barlow & Brunk (1972) had proved a functional relationship of the isotonic squared error minimization result to any other convex functions even for partial , demonstrating that the same solution holds for NLL and Brier score. In addition, the above stated theorem was previously established by Brummer and Perez (Brummer & Preez, 2013), a result that we were unaware of during the writing process. Nevertheless, we provide a much simpler and more accessible proof.

**Notations.** Assume w.l.o.g. that $x_i \lesssim x_j$ whenever $i < j$. Define:

- $\tilde{u}(i) = \{U \mid i \in U\}$: the set of possible upper sets for $i$.

- $\tilde{l}(i) = \{L \mid i \in L\}$: the set of possible lower sets for $i$.

- $\text{Avg}(S) = \frac{\sum_{k \in S} w_k y_k}{\sum_{k \in S} w_k}, S \subseteq \{1, \ldots, n\}$: The weighted average of $S$.

- $S_i^j = i \to j = \{k \mid i \leq k \leq j\}$: The set of indices from $i$ to $j$.

- $U(S) = \{U \mid U \subseteq S \wedge U \text{ is upper set in } S\}$: The set of all upper sets within a subset $S$.

two key properties of the described solution:

1. **The solution is isotonic:** Since for $i < j$, $\tilde{l}(j) \subset \tilde{l}(i)$ and $\tilde{u}(i) \subset \tilde{u}(j)$, we have:

$$\hat{g}(x_i) = \min_{\tilde{l}(i)} \max_{\tilde{u}(i)} \text{Avg}(\tilde{u}(i) \cap \tilde{l}(i)) \leq \min_{\tilde{l}(j)} \max_{\tilde{u}(j)} \text{Avg}(\tilde{u}(i) \cap \tilde{l}(j)) \leq \min_{\tilde{l}(j)} \max_{\tilde{u}(j)} \text{Avg}(\tilde{u}(j) \cap \tilde{l}(j)) = \hat{g}(x_j).$$

2. **Fixed on Partition:** The described solution creates a partition $I$ of $\{1, 2, \ldots, n\}$ such that for every $A \in I \subset \{1, 2, \ldots, n\}$, it holds that $\hat{g}(x_i) = \hat{g}(x_j)$ for all $i, j \in A$. Let $u_i^*$ and $l_i^*$ represent the respective maximizing upper and lower sets for the min-max solution, with $A = u_i^* \cap l_i^*$. For any $j \in A$, if $j < i$, we have $\tilde{u}(j) \subset \tilde{u}(i)$. Thus, since $j \in A, u_j^* = u_i^*$, leading to $\hat{g}(x_i) = \hat{g}(x_j)$. If $i < j$, we know that $\text{Avg}(A) \leq \text{Avg}(S_{\min(u_i^*)}^j)$ since we minimize over $\tilde{l}(i)$. Thus, $\text{Avg}(S_j^{\max(u_i^*)}) \leq \text{Avg}(A)$, leading to $\text{Avg}(S_{\min(u_i^*)}^n) \geq \text{Avg}(S_j^n)$ for all $j > i$. Therefore, $u_j^* = u_i^*$, which leads to $\hat{g}(x_i) = \hat{g}(x_j)$.

**Lemma D.2.** *Convexity lemma for binary proper scoring rules. For a binary proper scoring rule $S^*$, the minimizer over the interval $[\underline{p}, \overline{p}]$ is given by:*

$$\arg \min_{\underline{p} \leq q \leq \overline{p}} S^*(q, p) = \max(\underline{p}, \min(\overline{p}, p)).$$

*Proof.* By definition:

$$S^*(q, p) = pS(q, 1) + (1 - p)S(q, 0),$$

and using the properties of proper scoring rules (see Schervish (1989)), we have:

$$S(q, 1) = e(q) + (1 - q)e'(q), \quad S(q, 0) = e(q) - qe'(q),$$

where $e$ is a concave function. Thus:

$$S^*(q,p) = e(q) + (p-q)e'(q),$$

and the derivative of $S^*$ with respect to $q$ is:

$$\frac{\partial S^*(q,p)}{\partial q} = e'(q) + (p-q)e''(q) - e'(q) = (p-q)e''(q).$$

Since $e$ is concave, $e''$ is non-positive. Therefore, $S(q,p)$ is monotonic decreasing for $q \leq p$ and monotonic increasing for $q > p$. Thus, the minimum will be obtained at:

$$\max(\underline{p}, \min(\overline{p}, p)).$$

$\square$

**Lemma D.3.** *Let* $y_1, y_2, \ldots, y_n$ *be a series where* $y_i \in \{0, 1\}$, *with respective weights* $w_1, w_2, \ldots, w_n$. *Suppose the following property holds:*

$$(*) \quad \arg\max_U Avg(U) = \{1, 2, \ldots, n\} = [n]$$

*Then the bounded isotonic regression:*

$$\max_g \sum_{i=1}^n S(g(i), y_i)w_i \quad s.t. \quad g(i) \leq g(j) \, for \, i \leq j; \quad \underline{p} \leq g \leq \overline{p}$$

*is solved by:*

$$\hat{g} = \max(\underline{p}, \min(\overline{p}, Avg([n]))).$$

*Proof.* By induction. For $n = 1$, the solution follows directly from the properties of the scoring rule $S$, as $S(\bullet, 1)$ is monotonic increasing and $S(\bullet, 0)$ is monotonic decreasing. Assume the result holds for any series of length $\leq n$, and prove for $n + 1$.

Let $y_1, y_2, \ldots, y_n, y_{n+1}$ be a series holding $(*)$. We will assume $y_{n+1} = 0$, as otherwise the series is entirely composed of 1s, and the statement follows naturally. Let $k$ be the index of the maximal appearing sequence which holds $(*)$, meaning:

$$k = \max_{i \leq n} \left( \{i \mid \arg\max_{U \in U(S_1^i)} Avg(U) = [i]\} \right).$$

We will show that both $A_0 = \{y_1, y_2, \ldots, y_k\}$ and $A_1 = \{y_{k+1}, \ldots, y_{n+1}\}$ hold $(*)$. The set $A_0$ holds $(*)$ by its selection. For $A_1$, assume it does not hold. Then there exists $k + 1 < l \leq n + 1$ such that:

$$\arg\max_{U \in U(S_{k+1}^{n+1})} Avg(U) = S_l^{n+1},$$

yielding the following inequalities:

1. $Avg(S_l^{n+1}) \geq Avg(S_{k+1}^{n+1})$

2. $Avg(S_l^{n+1}) \geq Avg(S_{k+1}^{l-1})$

3. $Avg(S_l^{n+1}) \leq Avg(S_1^{n+1})$

4. $Avg(S_l^{n+1}) \leq Avg(S_1^{l-1})$.

Inequalities (3) and (4) hold because $\{y_1, \ldots, y_{n+1}\}$ satisfies $(*)$. Now, since $l - 1 > k$, there exists an index $1 < l' < l - 1$ such that:

$$S_{l'}^{l-1} = \arg \max_{U \in U(S_1^{l-1})} \text{Avg}(U),$$

and in particular, this implies:

$$(5) \quad \text{Avg}(S_1^{l-1}) \leq \text{Avg}(S_{l'}^{l-1}).$$

If $l' > k$, it would contradict the fact that $S_l^{n+1}$ is the maximal upper set for $A_1$. From (4) and (5), we have:

$$(6) \ \text{Avg}(S_{l'}^{n+1}) = \lambda \text{Avg}(S_{l'}^{l-1}) + (1 - \lambda)\text{Avg}(S_l^{n+1}) \geq \lambda \text{Avg}(S_1^{l-1}) + (1 - \lambda)\text{Avg}(S_l^{n+1}) \geq \text{Avg}(S_l^{n+1}),$$

thus $l' \leq k$. This allows us to rewrite (5) as:

$$(1 - w_1)\text{Avg}(S_{l'}^k) + w_1 \text{Avg}(S_{k+1}^{l-1}) \geq (1 - w_2)\text{Avg}(S_1^k) + w_2 \text{Avg}(S_{k+1}^{l-1})$$

where:

$$w_1 = \frac{\sum_{i=k+1}^{l-1} w_i}{\sum_{i=l'}^{l-1} w_i}, \quad w_2 = \frac{\sum_{i=k+1}^{l-1} w_i}{\sum_{i=1}^{l-1} w_i}.$$

Noting that $w_1 \geq w_2$ and the fact that $S_{l'}^k \geq S_{k+1}^l$, we obtain from (6):

$$(1 - w_1)\text{Avg}(S_{l'}^k) + (w_1 - w_2)\text{Avg}(S_{l'}^k) \geq (1 - w_2)\text{Avg}(S_1^k),$$

which implies $\text{Avg}(S_{l'}^k) \geq \text{Avg}(S_1^k)$, contradicting the assumption that $A_0$ holds $(*)$. Therefore, both $A_0$ and $A_1$ hold $(*)$, and we can apply our induction theorem to them.

marking $q_1, q_2$ for $\hat{g}|_{S_1^k}$ and $\hat{g}|_{S_{k+1}^{n+1}}$, respectively, we have:

$$q_2 = \begin{cases} \text{Avg}(S_{k+1}^{n+1}) & \text{if } q_1 \leq \text{Avg}(S_{k+1}^{n+1}) \leq \overline{p}, \\ q_1 & \text{if } \text{Avg}(S_{k+1}^{n+1}) \leq q_1 \leq \overline{p}, \\ \overline{p} & \text{if } q_1 \leq \overline{p} \leq \text{Avg}(S_{k+1}^{n+1}). \end{cases}$$

Using the convexity lemma for proper scoring rules, and since $\text{Avg}(S_{k+1}^{n+1}) \leq \text{Avg}(S_1^k)$ as $\{y_1, \ldots, y_{n+1}\}$ holds $(*)$, we get that if $q_2 > q_1$, we can further decrease our score by increasing $q_1$, leading to the conclusion that in the optimal solution $q_1 = q_2 = q^*$. Now:

$$\max_q \sum_{i=1}^{n+1} S(q, y_i)w_i = \max_q \sum_{i=1}^{n+1} \frac{S(q, y_i)w_i}{\sum_{j=1}^{n+1} w_j} = \max_q \frac{\sum_{j=1}^{n+1} w_j \cdot y_j}{\sum_{j=1}^{n+1} w_j} S(q, 1) + \frac{\sum_{j=1}^{n+1} w_j \cdot (1 - y_j)}{\sum_{j=1}^{n+1} w_j} S(q, 0).$$

Let $\tilde{p} = \frac{\sum_{j=1}^{n+1} w_j \cdot y_j}{\sum_{j=1}^{n+1} w_j}$. Then:

$$\max_q \tilde{p} S(q, 1) + (1 - \tilde{p})S(q, 0) = \sup_q S^*(q, \tilde{p}) = \max(\underline{p}, \min(\overline{p}, \tilde{p})),$$

leading to our desired claim. $\qquad\square$

From the convexity lemma and induction base cases, and with our two earlier observations, Theorem D.1 proof becomes straightforward.

*Proof.* Let $I$ be a partition of $\{1, 2, \ldots, n\}$ achieved by the described solution. We notice that every $A \in I$ holds the lemma property, as for every group, the maximization set for the given lower set is the same, including the last element in the group. Knowing that:

$$\min_g \sum_{i=1}^n S(g(x_i), y_i) w_i \geq \sum_{A \in I} \min_g \sum_{i \in A} S(g(x_i), y_i) w_i,$$

subject to:

$$g(x_i) \leq g(x_j) \text{ when } x_i \lesssim x_j,$$

and given the lemma, we know that the solution for the latter problem is:

$$\forall A \in I, \forall i \in A : \hat{g}(x_i) = \mathrm{Avg}(A).$$

Given the isotonic nature of this solution, it is also feasible for the first minimization problem, proving our desired result.

$\square$

