# OpenReview forum: "Improving Multi-Class Calibration through Normalization-Aware Isotonic Techniques"
_ICML.cc/2025/Conference — ICML 2025 poster_

### Official Review · Reviewer_Rvdp · 2025-03-14

**Overall Recommendation:** 3

**Summary:**

This paper addresses  extending isotonic regression from binary calibration to multi-class calibration. It proposes isotonic normalization-aware techniques for multi-class calibration. In particular, it introduces two techniques to account for probability normalization: (1) NA-FIR that incorporates normalization directly into the optimization process and (2) SCIR that models the problem as a cumulative bivariate isotonic regression. Experiments  are conducted on several text and image classification datasets across different model architectures, showing the proposed method can improve  log-likelihood (NLL) and expected calibration error (ECE) metrics.



## update after rebuttal:
The experiments on ImageNet-1k During rebuttal indeed drive me towards positive, but not upon an accept. Even though my concerns on time cost still holds, the explanation (in particular the second round) indeed drive me to positive. I decide to raise my score from 2 (weak reject) to 3 (weak accept).

**Claims And Evidence:**

The main claim of this paper is from the empirical perspective tha "our proposed methods consistently improve both negative log-likelihood (NLL) and calibration error across diverse datasets, achieving SOTA results and reaffirming the effectiveness of IR for calibration". While this claim is indeed supported by the experiments of the paper, I have concerns on the experiments, relating to the scalability, see the details of the comments on experimental designs.

**Essential References Not Discussed:**

I believe this paper provides essential references, even though I am not familiar to the references about calibration.

**Experimental Designs Or Analyses:**

Even thought the experiments support the claims, I still have concerns on the significance. My concerns are follows:

(1) The datasets used in this paper is almost small-scale, I think this paper should conduct experiments on more larger datasets, e.g., at least conduct experiments on ImageNet-1000 datasets (there are many pre-trained models for Imagenet-1000, I do not think the computing power is the problem).

(2) This paper should conduct experiments to compare the computation cost, e.g., wall clock time, especially on the Imagenet-1000 datasets. It is important show its scalability in practice. I think TS is a very simple (efficient) and effective method for calibration (even though I am not an experts in calibration),  How does the proposed method compared to TS in efficiency?

**Methods And Evaluation Criteria:**

The proposed method and evaluation criteria is overall make sense. But it misses the experiments in comparing the efficiency of the proposed methods and the benchmark is overall small-scale (see comments on experimental designs)

**Other Comments Or Suggestions:**

(1) Line 124, missing "."

(2) eqn 2, what is the meaning of "$\preceq$" ?

(3) Line 205 (right) , missing "," before "we justify...."

**Other Strengths And Weaknesses:**

NA

**Questions For Authors:**

How does the proposed method compared to TS in efficiency? especially with the increase on the size of classes and data samples? (better provide the results on ImageNet-1000)

**Relation To Broader Scientific Literature:**

This paper provides clear routing how this paper builds on. I acknowledge it. However, it seems incremental overall, from the perspective of methods.

**Theoretical Claims:**

I do not think this paper provides theoretical contributions. The method is based on previous work with some intuition, and the main technical contributions are the proposed algorithms.

---

> ### Author Rebuttal · Authors · 2025-04-01
>
> Thank you for your thoughtful review.
>
> As you rightfully pointed out, validating our methods on larger datasets such as ImageNet-1K is important for demonstrating scalability and generality. In the last few days we have made a concentrated effort and conducted additional experiments on ImageNet-1K using [eight pre-trained models](https://github.com/huggingface/pytorch-image-models/blob/main/results/results-imagenet.csv), using the 50k validation for both calibration (25%) and test (75%).
>
> We show here preliminary results for the calibration measures used in the paper. The results are in the same format as Table 2 in the paper, and show average rank and % best. As can be seen, for all measures except cw-ECE one of our methods NA-FIR and SCIR is best, in most cases by far. cw-ECE is not really a relevant measure with 1000 classes (see paper for definition of cw-ECE), as a vector of all zeros is likely to be best.
>
> | Metric            | NA-FIR      | SCIR        | FIR         | IR OvR      | TS          | VS          | MS          | Uncalibrated |
> |------------------|-------------|-------------|-------------|-------------|-------------|-------------|-------------|---------------|
> | **Cross Entropy (NLL)** | **1.0 (100%)**  | 4.9 (0%)    | 2.0 (0%)    | 8.0 (0%)    | 3.4 (0%)    | 5.1 (0%)    | 5.9 (0%)    | 5.8 (0%)      |
> | **Brier Score (BS)**   | **1.2 (75%)**   | 3.4 (0%)    | 2.4 (0%)    | 6.5 (0%)    | 4.1 (12%)   | 5.8 (12%)   | 6.6 (0%)    | 6.0 (0%)      |
> | **conf-ECE**| **1.4 (75%)**   | 2.0 (25%)   | 3.0 (0%)    | 5.5 (0%)    | 4.0 (0%)    | 6.1 (0%)    | 7.4 (0%)    | 6.6 (0%)      |
> | **cw-ECE**          | 3.6 (0%)    | 6.1 (0%)    | 3.1 (12%)   | 4.4 (0%)    | **2.0 (50%)**   | 4.1 (25%)   | 5.4 (12%)   | 7.2 (0%)      |
> | **TECE**          | 2.9 (12%)   | **2.1 (50%)**   | 2.5 (0%)    | 8.0 (0%)    | 5.1 (0%)    | 6.4 (0%)    | 6.1 (0%)    | 2.9 (38%)     |
>
>
> Regarding runtime, we acknowledge that our methods are **less efficient** than TS or FIR on large datasets as you justly were concerned.
> While both TS and FIR run in a few seconds, for the algorithms presented in the paper it takes a mean time of 120 minutes for SCIR and 16 minutes for NA-FIR with the suggested algorithm.
>
> We would like to add additional context regarding these results:
> * SCIR has a theoretical worst case time complexity of $O(m^2k^4)$ (Appendix A), and though it scales better in practice, it remains resource-intensive for 𝑘=1000 classes. For example, fitting takes around 2 minutes for cifar100 (100 classes, 5k calibration samples - see Appendix C) and for imagenet-1k it takes around 120 minutes with 30 minutes std.
> * NA-FIR scales more efficiently even as k grows, since it fits a PAVA step first, significantly reducing the number of effective parameters. After applying memorization to only update bins deltas in NLL calculation, NA-FIR  for imagenet-1k (1000 classes) takes ~15 minute, in comparison to 5 minutes for food101 dataset (101 classes, 6.3k calibration samples). We also discuss several ways to optimize NA-FIR in the supplemental material which can result in significant time difference (using this approach results in about a minute fitting time, but results were less stable).
>
> While we recognize that TS remains the most efficient method and is a strong baseline, our approach offers stronger calibration under normalization constraints not addressed by previous nonparametric methods. We provide both theoretical grounding (see Section D and our response to Reviewer 1) and empirical evidence including on large scale dataset as Imagent-1k to support this claim.
>
> Finally, in many applications, calibration sets are limited (see for example [1]), and compute overhead from calibration is minor compared to model training costs. If accepted we will include wall-clock details in the final version for transparency, thus by offering complexity analysis and strong empirical gains, we allow practitioners to weigh the pros and cons depending on their application and compute resources. We believe our methods have a place in the literature, as they offer flexible and effective alternative where calibration quality is prioritized.
>
> We also appreciate the note on typos and will revise accordingly.  Regarding the use of $\preceq$, we follow convention where it denotes partial ordering.
>
> [1] Patel et al, Multi-class uncertainty calibration via mutual information maximization-based binning, ICLR 2021

---

> > ### Comment · Reviewer_Rvdp · 2025-04-07
> >
> > Thanks for the response to my comments.  The authors provide the results on ImageNet-1K during rebuttal. Even though the proposed method obtain overall better performances (calibration metrics) than the baselines, I still have concerns on its scalability, due to its low efficiency.  It is clear the proposed method has high computation complexity, e.g. , "While both TS and FIR run in **a few seconds**, for the algorithms presented in the paper it takes a mean time of **120 minutes** for SCIR and **16 minutes** for NA-FIR with the suggested algorithm." (It is good the authors provide the results in the rebuttal, but this results should be provided in the submission.).  Besides, I am not comfort to the respond that " **If accepted** **we will include wall-clock details** in the final version for transparency, thus by offering complexity analysis and strong empirical gains". I think the submission should include the wall-clock details, no matter whether the paper is accepted. Clearly, the main contribution of this paper is the proposed method , and the authors should provide a comprehensive comparisons, including effects and efficiency (no matter merits and drawbacks).
> >
> > Overall, the proposed method is not better enough, compared to the TS method (a method proposed in 2017), by trade-offing the performance and the efficiency.  Besides, I do not think this paper is a theoretical paper (I do not recognize its theoretical contribution). I thus keep negative to this paper currently.

---

> > > ### Author Response · Authors · 2025-04-09
> > >
> > > Thank you for your response.
> > >
> > > Regarding the phrase "If accepted," our intention was simply to indicate that submissions cannot be modified unless they are accepted. As for the ImageNet-1K results, they were included in the rebuttal since the experiments were conducted during the rebuttal phase. Additionally, as previously mentioned, reporting wall-clock time is not standard practice in the calibration literature—in fact, to our knowledge, no prior work has reported these details.
> > >
> > > As the reviewer notes, our main contribution lies in the two proposed non-parametric methods, which are grounded in well-articulated intuitions, have clear computational complexity, and are supported by a comprehensive experimental design. We believe that the information provided enables any practitioner to fairly evaluate the strengths and limitations of our approach—and, as the reviewer has effectively illustrated through their own reasoning, to decide whether the additional computation is worthwhile for improved calibration (which, we note, remains negligible compared to training most models in this area)

---

### Official Review · Reviewer_6BKQ · 2025-03-15

**Overall Recommendation:** 4

**Summary:**

The paper addresses the critical challenge of multi-class calibration in supervised learning.
While isotonic regression has proven effective for binary calibration problems, its extension to multi-class settings through one-vs-rest (OvR) calibration has historically underperformed compared to parametric methods like Temperature Scaling.
The authors identify a key limitation: traditional approaches do not inherently account for probability normalization during optimization.

The primary contributions are two novel isotonic regression techniques for multi-class calibration:
1.    Normalization-Aware Flattened Isotonic Regression (NA-FIR): This method explicitly incorporates normalization into the optimization process, modifying the standard isotonic regression approach.
 2.   Cumulative Bivariate Isotonic Regression (SCIR): Rather than treating each class as an independent binary task, this approach redefines the problem by addressing cumulative sub-problems.

The authors demonstrate that these normalization-aware techniques consistently improve calibration performance across diverse datasets and model architectures, asserting that their approach represents a state-of-the-art non-parametric alternative in scenarios where parametric assumptions may be limiting.
The paper presents empirical evaluations on text and image classification datasets, demonstrating that the proposed approach improves negative log-likelihood (NLL) and expected calibration error (ECE) metrics.

## update after rebuttal
I keep my score - the empirical results are enough IMHO, even though *some* of them are on rather small datasets.

**Claims And Evidence:**

The main claim of the paper is that:
>  NA-FIR and SCIR (the proposed methods) improve multi-class calibration by inherently accounting for probability normalization.

To which the autors provide the following empirical evidence
* Figure 1 visually supports this claim by showing improved calibration with NA-FIR and SCIR.
* Section 5 that show improvements in NLL and ECE, which are standard metrics for evaluating calibration.
* The results are presented in Table 2, where the proposed methods generally achieve better average rankings compared to other calibration techniques.

The evidence provided includes empirical results from experiments on six benchmark datasets (CIFAR-10, CIFAR-100, Food-101, R52, NG20, and Yelp Review) using various modern deep neural network architectures and classical machine learning models.

**Essential References Not Discussed:**

I could not think of a major missing reference.

**Experimental Designs Or Analyses:**

The experimental section is following the accepted practice in the field, as I mention above.

I would commend the authors for using cross-validation (line 867) which is sadly not a common practice (IMHO)

**Methods And Evaluation Criteria:**

NLL and ECE are two of the most accepted ways to measure calibration in machine learning.
The authors also employ variations of ECE, such as class-wise ECE (cw-ECE), thresholded equal-mass binning ECE (TECE), and confidence-calibrated ECE (conf-ECE).
The authors use traditional benchmark datasets from both image and text domains, and compare methods against several existing calibration techniques (TS, MS, VS, IR-OvR, FIR).

**Other Comments Or Suggestions:**

.

**Other Strengths And Weaknesses:**

.

**Questions For Authors:**

.

**Relation To Broader Scientific Literature:**

The paper effectively situates itself within the broader calibration literature.
The authors acknowledge foundational works on calibration by Murphy (1973), Murphy & Winkler (1977), and DeGroot & Fienberg (1981), establishing historical context.

More recent developments are referenced, including Platt's (1999) logistic regression-based calibration for binary predictions and subsequent extensions by Kull et al. (2017).
For neural networks specifically, the authors discuss Guo et al.'s (2017) parametric methods: Matrix Scaling, Vector Scaling, and Temperature Scaling.

In the non-parametric domain, the paper builds upon Zadrozny & Elkan's (2002) work on decomposing multi-class problems into binary subproblems and more recent contributions by Patel et al. (2020) and Zhang et al. (2020) on Flattened Isotonic Regression.

The authors identify a gap in the literature where prior work reported state-of-the-art results for unnormalized predictions, which they argue conflicts with the fundamental goal of calibration: fostering trust in reported probabilities.

**Theoretical Claims:**

The paper includes a theoretical claim in *Proposition 4.1*, which states that the cumulative sorted problem defined in Eq. (5) can be solved in $O(m^2k^4)$ using *Algorithm 2*.

The proof for the proposition, which I could not follow, is provided in Appendix A.

---

> ### Author Rebuttal · Authors · 2025-04-01
>
> Thank you for your thoughtful and positive review.
> As far as theoretical claims we would like to also refer the reviewer to Appendix 4 where we provide a theoretically valid motivation for our isotonic approach.

---

### Official Review · Reviewer_mZju · 2025-03-17

**Overall Recommendation:** 4

**Summary:**

he paper discusses a tweak on a post-hoc recalibration method for the multi-class classification algorithm. One of the widely used methods for post-hoc recalibration in the binary setting is the isotonic regression: given already a classification algorithm, we can consider a new regression problem where the covariates $x_i$ are now the predictions $p_i$ of the previous algorithms, and the response variables $y_i \in \{0, 1\}$ are associated with the actual distributions. The isotonic regression finds a recalibration function $g : [0, 1] \to [0,1]$ minimizing a proper-scoring loss $\mathbb{E}_i \ell(g(p), y)$ among all non-decreasing functions $g: [0,1] \to [0,1]$. As it turns out, given sample $(p_1, y_1), \ldots (p_n, y_n)$, there exist an efficient algorithm solving this minimization problem exactly, and the solution is always a piecewise constant function.

The generalization of this recalibration method to multi-class classification setting is much more tricky; now, the algorithm outputs a probability vector $p \in \Delta^k$, and we would like to provide some re-parametrization of $p$ that improves the proper loss. A method proposed previously (Patel et al. 202, Zhang et al. 2020) is to just attempt to minimize over all univariate monotone functiosn $g$ the negative log likelihood loss
$\mathbb{E} \sum_i - y_i \log( g(p(x)\_i))$, and then report the normalized $g(p\_i)/\sum\_{j} g(p\_j)$ as the final probability vector.

The authos observe that in the objective function here, the log-likelihood is calculated with respect to a vector $g(p_i)$ over $i \in [k]$ which is not necessairly contained in the probability simplex. They suggest directly minimize over monotone $g$ the desired log-likelihood
$\mathbb{E} \sum_i - y_i \log(g(p(x)_i) / \sum_j g(p(x))_j)$.


This is relatively simple and natural tweak to the idea of Patel et al and Zhang et al.  Since the empirical minimization problem is not convex anymore, in contrast with the simple one-dimensional situation where the global minimum could be obtained explicitly, here it is not clear how to actually find a reasonable function $g$.

The authors tried several optimization heuristics to solve the computational problem at hand, and report the best of the attempted solutions. They ended up starting with a piecewise-constant function provided by a standard one-dimensional isotonic regression for the unnormalized problem, splitting the regions on which the function is constant into shorter sub-intervals, and applying a simulated annealing to adjust the values in each interval.

**Claims And Evidence:**

The authors claim that their modification of the known recalibration method FIR, despite looking relatively innocent, provides a significant adventage and results in noticably better calibration than state of the art alternatives. They compared several post-hoc calibration methods on six benchmarks, with varying number of classes. For each pair of recalibration methods they showed on a heat-map percentage of the times one method achieved a better score than the other. The newly proposed NA-FIR and SCIR seems to consistently outperform other six calibration methods.

**Essential References Not Discussed:**

I am not aware of the essential references that are missing.

**Experimental Designs Or Analyses:**

Experimental design seems sound, and --- importantly, it is using a wide range of datasets coming from different sources. Specific details of all experiments, and additional metrics of calibration errors can be found in the extensive supplementary material. I did not attempt to replicate their experiments.

**Methods And Evaluation Criteria:**

The author has used quite a wide set of benchmark datasets, and compared a wide range of different recalibration algorithms. They also attempted to use various heuristic to solve the specific non-convex optimization problem they used in the definition of NA-FIR~---they only present the final results for the best heuristic, but it is valuable that they mention other attempts they tried.

**Other Comments Or Suggestions:**

-

**Other Strengths And Weaknesses:**

The experimental setup in this paper is very well prepared, with a wide range of data sets, wide range of other recalibration algorithms used as comparisons, and few calibration metrics to compare those algorithms. Moreover, the supplementary material providing very specific details of how the experiments were preformed, making it quite easy to attempt potential replication.

**Questions For Authors:**

-

**Relation To Broader Scientific Literature:**

The problems of multi-class recalibration, and in general of understanding various aspects multi-class calibration are important areas with still a large room for further discoveries. The authors propose an improvement over known methods for recalibration, while applying the isotonic regression --- preserving the monotonicity of the post-processing function $g$.

**Theoretical Claims:**

There are no theoretical claims in this paper~---they provide experimental evidence for the validity of their approach.

---

> ### Author Rebuttal · Authors · 2025-04-01
>
> Thank you for your thoughtful and positive review.
> The reviewer rightfully mentions that only final results for the best heuristic are being presented,  but we refer to our Appendix C and comments to Reviewer 4 where we provide more details on computational considerations for adopting our suggested methods that will be added to our final version if accepted.

---

### Official Review · Reviewer_7kNs · 2025-03-19

**Overall Recommendation:** 2

**Summary:**

This paper proposes two isotonic regression based approaches that incorporate normalization into problem formulation of multiclass calibration:
* Normalized Aware Flattened Isotonic Regression, which finds a mapping g such that g(p(x)) is normalized before computing the NLL objective.
* Sorted Cumulative Isotonic Regression, which finds a g that minimizes binary cross entropy loss on some set.

**Claims And Evidence:**

The paper proposes two calibration methods with builtin normalization. Those methods can ensure normalized output, but my major concern is the statistical consistency. That is, assuming the assumptions are correct for the underlying distribution, there is no guarantee that the proposed algorithms will achieve the meaningful optima as measured by some proper scoring rule (or even any weaker metrics such as different forms of calibration errors). For the first method, the authors point out that it may not recover global optima because of the nonconvexity. For the second method, the algorithmic convergence is given but the statistical convergence or consistency is missing.

**Essential References Not Discussed:**

* Both major assumptions discussed in Section 3 have been studied before [1]. Using the concept introduced in [1], Category Independence means p \mapsto P[Y|\hat{p}(X)=p] is diagonal, and Order Preserving means \hat{g} is intra-order preserving.
* Moreover, I think Section 4 needs improvement in terms of conciseness, clarity, and organization. [1] did a better job in terms of being precise with the language.

1. Rahimi et al., Intra Order-Preserving Functions for Calibration of Multi-Class Neural Networks, NeurIPS 2020

**Experimental Designs Or Analyses:**

No.

**Methods And Evaluation Criteria:**

The first approach: The intuition make sense. The assumptions are strong and needs more technical discussion. See my last point in the question section.

The second approach: I find it difficult to parse the equations. For example, in Equation (5), \tilde{y} is not defined; function g is not defined (why does it take two arguments?). As such, I'm not sure if the intuition checks out for this approach.

**Other Comments Or Suggestions:**

Typo:
* There's an extra "2025" in the running title at the top of each page.
* L037 right: "have highlighted the effectiveness of ”simple” para-
metric approaches": please fix the left quotation mark.
- L268 left: Use macro \log (so that it's not displayed in italics). Use \left( and \right) around the fraction as in L273.
* L273 left: NA-FIR should be wrapped in \text{} as in L268.

**Other Strengths And Weaknesses:**

Strength: the experiments cover many datasets.

**Questions For Authors:**

- L189 right: "We believe valid probability predictions are essential for building trust among practitioners, especially in decision-making frameworks, thus the only alternative offered is assuming Category Independence."
	- Can you elaborate the logic implied by "thus"? I understand that ensuring valid probability predictions is equivalent to requiring the calibration function is a mapping from the simplex to itself; why does it necessarily require the Category Independence assumption to hold?
	- Also this paragraph seems out of place. The "This" at the beginning of the following paragraph refers to the weakness in the previous paragraph, and I'm not clear of the goal of this paragraph.
- L197 right: "...differences that cannot be solely attributed to observed miscalibration in the intervals around the marginal predictions of 0, 0.1, 0.2."
	- I don't understand what this means.
	- It is well known that marginal calibration (do you mean classwise-calibration in Def 2.2?) is strictly weaker than (canonical) calibration (Def 2.1). What is the point of using an example here to illustrate this?
- L268 left: Equation (4) is a sample version of NLL. I am generally skeptical about breaking down the classes in the objective function which requires strong assumptions. Summing over the classes in Equation (4) essentially treats every class dimension of every sample is independent. This is in general not true. However, I am aware that many prior works (most notably, see Step 1 Data Ensemble of Zhang'20 mix-and-match) take this assumption. My main concern is why minimizing such a loss function will optimize the proper scoring rule, or even any weaker metrics like canonical or classwise calibration error?

**Relation To Broader Scientific Literature:**

N/A

**Theoretical Claims:**

The authors discussed the algorithmic convergence of the second algorithm. I did not check the correctness because my major concern is in statistical concistency/convergence.

---

> ### Author Rebuttal · Authors · 2025-04-01
>
> Thank you for your thoughtful review.
>
> The major concern is regarding statistical properties of the algorithms we propose, in particular NA-FIR. This is expressed in the review sections on Claims and Evidence, Theoretical Claims and the last point of Questions for Authors.
>
> Regarding the statistical properties, it is important to emphasize that the NA-FIR loss function in Eq. (4) is a standard multinomial NLL (with one expression for each observation), in contrast to the loss function for previous FIR variants (Eq. (3) in our paper) which is indeed “summing over the classes” as the reviewer notes. The first implication is that the reviewer’s last comment in Questions for Authors regarding Eq. (4) is incorrect since Eq. (4) is a standard multinomial NLL. It also means that NA-FIR indeed uses a proper scoring rule for calibration (it is well known since Good (1952) that multinomial NLL or “log score” is proper), which addresses the reviewer’s statistical consistency issue. NA-FIR is non-convex (as are other calibration methods), so does not guarantee algorithmic convergence, however, the loss function is proper and statistically consistent. We will make this point more explicit in the paper’s published version, if accepted. For our second algorithm SCIR in Eq. (5), indeed the loss function is similar to FIR and “sums over the classes”. We agree that this is not ideal in terms of statistical performance, but it is common practice, as the reviewer notes.
>
> Additional issues:
>
> * Clarity of the SCIR (second) approach, referred in second comment under Methods and Evaluation Criteria. We emphasize that all terms in Eq. (5) are defined, although we acknowledge that the definitions can (and should) be made clearer, especially given the conceptual complexity of the SCIR approach. Specifically, $\tilde{y}$ is the cumulative label, defined via the definition of $CU_{sorted}$ and explained right after Eq. (5). The function $g$ is the optimization objective (over 2d space of cumulative probability and rank), and is implicitly defined through that.
>
> * The first two Questions for Authors note clarity issues in the beginning of Section 4, when motivating the desired properties of multi-class calibration. We agree that the discussion in this part should be improved. Specifically to the reviewer’s comments, the word “thus” was intended to refer to other non-parametric approaches that assume category independence during calibration, and  normalize predictions in the actual prediction process. The example in l197 was indeed intended to demonstrate that marginal calibration is weaker than calibration in Def. 2.1 as the reviewer notes. This was intended for readers who may not be as familiar with these notions. We will make an effort to improve clarity of this part.
>
> * Missing reference to Rahimi et al. (2020) who define similar criteria to the ones we do in Sec. 3.3. Thank you for this reference, which is indeed relevant, and will be added to our paper and properly referenced. We note that our focus is on the algorithmic non-parametric contributions we propose, which are not related to their paper.
>
> * Typos noted under Comments and Suggestions. Thank you for these, we will address them.

---

### Decision · Program_Chairs · 2025-05-01

**Decision:**

Accept (poster)

**Comment:**

The paper proposes two normalization‑aware isotonic calibration methods for the multi‑class setting and supplies broad empirical evidence (since rebuttal including also ImageNet‑1K), that these techniques improve negative log‑likelihood and several calibration metrics over existing post‑hoc approaches. Two reviewers find the contribution sufficient for ICML and a third moved from weak reject to weak accept after seeing the large‑scale results, so the overall tilt of the reviews is positive. The fourth reviewer kept the score as weak reject but is now more neutral to its acceptance.

I highlight three key points in the discussions. First, one reviewer still reports difficulty following the notation and the rationale for sharing a single isotonic function across classes, so the exposition must be tightened and definitions made explicit. Second, the authors provided wall‑clock times that confirm NA‑FIR and especially SCIR are much slower than Temperature Scaling, and one reviewer continues to question whether the accuracy gains justify this computational cost; thus, the final version should present these timings and discuss use‑case trade‑offs. Third, the motivation and statistical justification for the shared isotonic assumption need clearer explanation to remove the remaining doubts about consistency. The new ImageNet‑1K experiments addressed the dataset‑scale concern, and no reviewer raised further objections. Provided the authors commit to clarifying the presentation and to documenting runtime, I recommend acceptance.